# POTENTIAL BASED DIFFUSION MOTION PLANNING

## ABSTRACT

Effective motion planning in high dimensional spaces is a long-standing open problem in robotics. One class of traditional motion planning algorithms corresponds to potential-based motion planning. An advantage of potential based motion planning is composability – different motion constraints can easily combined by adding corresponding potentials. However, constructing motion paths from potentials requires solving a global optimization across configuration space potential landscape, which is often prone to local minima, causing these approaches to fall out of favor in recent years. We propose a new approach towards learning potential based motion planning, where we train a neural networks to capture and learn an easily optimizable potentials over motion planning trajectories. We illustrate the effectiveness of such approach, significantly outperforming both classical and recent learned motion planning approaches, and illustrate its inherent composability, enabling us to generalize to a multitude of different motion constraints.

## 1 INTRODUCTION

Motion planning is a fundamental problem in robotics and aims to find a smooth, collision free path between a start and goal state given a specified configuration space, and is heavily used across a variety of different robotics tasks such as manipulation or navigation (Laumond et al., 1998). A variety of approaches exist for motion planning, ranging from classical sampling based approaches (Karaman & Frazzoli, 2011; Gammell et al., 2015; Kavraki et al., 1996; Kuffner & LaValle, 2000) and optimization based methods (Ratliff et al., 2009; Mukadam et al., 2018; Kalakrishnan et al., 2011). A recent body of works have further explored how learned neural networks can be integrated with motion planning for accelerated performance (Fishman et al., 2023; Yamada et al., 2023; Qureshi et al., 2019; Le et al., 2023).

A classical approach towards motion planning is potential based motion planning (Koren et al., 1991; Ratliff et al., 2009; 2018; Xie et al., 2020), where both obstacles and goals define energy potentials through which trajectories are optimized to reach. A great advantage of potential based motion planning is that different constraints to motion planning can be converted into equivalent energy potentials and directly combined to optimize for motion plans. However, such approach generates motion plans primarily based on the local geometry with greedy optimization, resulting in the long-standing local minima issues (LaValle, 2006). In addition, it typically requires implicit obstacle representations, which is hard to obtain in real-world settings.

We present a potential based motion planning approach leveraging diffusion models (Sohl-Dickstein et al., 2015; Ho et al., 2020) where diffusion models are used to parameterize and learn potential landscapes across configuration space trajectories between start and goal states. Our method maps the start state, goal state, and environment geometry directly into a learned latent potential space, eliminating the need to design sophisticated potential functions. These potential functions are fit directly over long-horizon plans, helping avoid local energy minima. Furthermore, the inherent stochasticity in diffusion model enables a more robust optimization and can generate diverse motion plans for a specific problem, enabling failure recovery. In addition, guided by both local and global environment geometry in learned potentials, our method provides faster planning and requires less collision checking, compared with problem-independent sampling-based planners.

One major hurdle of learning-based motion planners (Ichter & Pavone, 2019; Qureshi et al., 2019; Fishman et al., 2023) is their generalizability to unseen, more complex constraints. For example, models trained on sparse obstacles usually fall short of the scenarios with cluttered obstacles. By contrast, similar to prior potential based motion planning methods, our learned potentials can be additively composed together to jointly solve motion planning problems with sets of constraints. As illustrated in Figure 1, combining two potentials from different diffusion models enables us to opti-

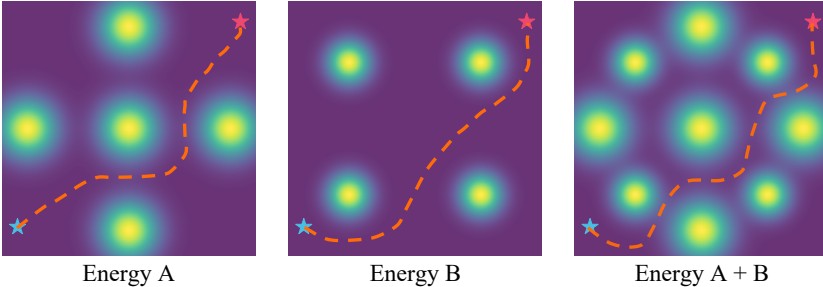

| Energy A | Energy B | Energy A + B |

Figure 1: **Illustrative Example of Composing Diffusion Energy Potentials.** Our approach learns different potential functions over motion planning trajectories (orange dashed lines) $q_{1:N}$. Different potentials can be combined and optimized to construct new motion plans that avoid obstacles encoded in both potential functions.

mize for trajectories that satisfy both constraints, one to avoid obstacles in a cross, and a second to avoid obstacles in a square. Such flexibility to ad-hoc composition of constraints is especially useful in robotics where agents will often experience new sets of motion constraints in its environment over the course of execution.

In addition to being able to combining different motion constraints together, we can also compose multiple instance of the sample diffusion potential together. This form of composition enables us to naturally generalize at inference time to motion planning problems with a larger number of obstacles than what have been observed at training time, by composing multiple instances of the learn diffusion obstacle potential model conditioned on subsets of the larger set of obstacles. We illustrate the effectiveness of such approach, substantially outperforming both classical and learned baselines.

Overall, in this paper, our contributions are three-fold. **(1)** We present an approach to learned potential based motion planning using diffusion models. **(2)** We illustrate the effectiveness of our approach, outperforming existing classical and learned motion planning algorithms. **(3)** We illustrate the compositionality of motion planner, enabling it to generalize to multiple sets of motion constraints as well as an increased number of objects.

## 2 RELATED WORK

**Motion Planning.** Classic sampling-based motion planners (Kavraki et al., 1996; Kuffner & LaValle, 2000; Elbanhawi & Simic, 2014; Gammell et al., 2014; Janson et al., 2015; Choudhury et al., 2016; Strub & Gammell, 2020) have gained wide adoption due to their completeness and generalizability. However, problem-independent nature of these methods can result in inefficiency particularly when planning for similar problems repetitively. Reactive methods, such as potential-based approaches (Khatib, 1986; Ratliff et al., 2018; Xie et al., 2020), velocity obstacles (Fiorini & Shiller, 1998; Van den Berg et al., 2008), and safety barrier certificates (Wang et al., 2017) can provide fast updates and have the guarantee for obstacle avoidance. However, their performance is typically constrained by local minima or numerical instability issues (LaValle, 2006), and they usually need to construct obstacle representations in the robot configuration space, which is hard to obtain especially in high dimension. To address these issues, recent works have proposed many deep-learning based motion planners (Ichter & Pavone, 2019; Qureshi et al., 2019; Bency et al., 2019; Fishman et al., 2023). These methods can generally increase planning speed, expand the planning horizon, or reduce the access queries to the environment by leveraging learned knowledge. One important line of research is combining neural network with sampling-based methods (Johnson et al., 2021; Yu & Gao, 2021; Lawson & Qureshi, 2022), termed hybrid motion planner. Particularly, latest work (Saha et al., 2023; Carvalho et al., 2023) adapts diffusion model as an auxiliary prior for trajectory generation, but still require accurate ground-truth cost function and dense environment queries when planning. In addition, many existing methods are only constrained to simple 2D environments (Yonetani et al., 2021; Chaplot et al., 2021; Toma et al., 2021). Contrary to them, we propose a motion planner applicable to various environments with different dimensionality while with shorter planning time and notably less environment access (i.e., collision checks). In addition, our potential formulation also equips our model with high generalization capability to out-of-distribution environment.

**Diffusion Models for Robotics.** Many recent works have explored the application of diffusion model for robotics (Janner et al., 2022; Chen et al., 2022; Kapelyukh et al., 2023; Ha et al., 2023). Current research spans a variety of robotics problems, including action sequence generation (Liang et al., 2023; Fang et al., 2023; Li et al., 2023), policy (Wang et al., 2023; Kang et al., 2023),

grasping (Urain et al., 2023; Huang et al., 2023), and visuomotor planning or control (Dalal et al., 2023; Yang et al., 2023a; Chi et al., 2023), with recent work also exploring their application in solving manipulation constraints (Yang et al., 2023b). In contrast to these works, we focus on how diffusion models can be used to explicitly parameterize and learn potentials in potential based motion planning. We illustrate the efficacy of such an approach and its ability to compose with other learned potentials.

## 3 METHOD

In this section, we first introduce potential based motion planning in Section 3.1. We then discuss how potential based motion planning can be implemented with diffusion models in Section 3.2. We further discuss how such an approach enables us to combine multiple different potentials together in Section 3.3. Finally, we discuss how we can refine motion plans generated by diffusion models in cases of collision in Section 3.4.

### 3.1 POTENTIAL BASED MOTION PLANNING

Given a specified start state $q_{\text{start}}$ and end state $q_{\text{end}}$ in a configuration space $\mathbb{R}^n$, motion planning is formulated as finding a collision-free trajectory $q_{1:N}$ which starts from $q_{\text{start}}$ and ends at $q_{\text{end}}$. To solve for such a collision-free trajectory $q_{1:N}$ in potential based motion planning (Koren et al., 1991), a potential function $U(q) : \mathbb{R}^n \to \mathbb{R}$ on the configuration space composed of

$$U(q) = U_{\text{att}}(q) + U_{\text{repel}}(q), \tag{1}$$

is defined, where $u(q)$ assigns low potential value to the goal state $q_{\text{end}}$ and high potential to all states which are in collision. In Equation 1, $U_{\text{att}}(q)$ represents a attraction potential that has low values at the the end state $q_{\text{end}}$ and high values away from it and $U_{\text{repel}}(q)$ represents a repulsion potential that has high values near obstacles and low values away them. The functional form of the potential function $U(q)$ provides an easy approach to integrate additional obstacles in motion planning by adding the a new potential $U_{\text{new}}(q)$ representing obstacles to the existing potential in Equation 1.

To obtain a motion plan from a potential field $U(q)$, a collision-free trajectory $q_{1:N}$ from $q_{\text{start}}$ to $q_{\text{end}}$ is obtained by iteratively following gradient of the potential function

$$q_t = q_{t-1} - \gamma \nabla_q U(q), \tag{2}$$

with a successful motion plan constructed when the optimization procedure reaches the minimum of the potential function $U(q)$. A major limitation of above approach in Equation 2 is *local minima* – if the optimization procedure falls in such a minima, the motion plan will no longer successfully construct paths from $q_{\text{start}}$ to $q_{\text{end}}$ (Yun & Tan, 1997; Teli & Wani, 2021).

### 3.2 POTENTIAL BASED DIFFUSION MOTION PLANNING

We next discuss how to learn potentials for potential motion planning that enable us to effectively optimize samples. Given a motion plan $q_{1:T}$ from start state $q_{\text{start}}$ to end state $q_{\text{end}}$ and a characterization of the configuration space $C$ (*i.e.* the set of obstacles in the environment), we propose to learn a trajectory-level potential function $U_\theta$ so that

$$q_{1:T}^* = \arg\min_{q_{1:T}} U_\theta(q_{1:T}, q_{\text{start}}, q_{\text{end}}, C), \tag{3}$$

where $q_{1:T}^*$ is a successful motion plan from $q_{\text{start}}$ to $q_{\text{end}}$.

To learn the potential function in Equation 3, we propose to learn a EBM (LeCun et al., 2006; Du & Mordatch, 2019) across a dataset of solved motion planning $D = \{q_{\text{start}}^i, q_{\text{end}}^i, q_{1:T}^i, C^i\}$, where $e^{-E_\theta(q_{1:T}|q_{\text{start}}, q_{\text{end}}, C)} \propto p(q_{1:T}|q_{\text{start}}, q_{\text{end}}, C)$. Since the dataset $D$ is of solved motion planning problems, the learned energy function $E_\theta$ will have minimal energy at successful motion plans $q_{1:T}^*$ and thus satisfy our potential function $U_\theta$ in Equation 3.

To learn the EBM landscape that enables us to effectively optimize and generate motion plans $q_{1:T}^*$, we propose to shape the energy landscape using denoising diffusion training objective (Sohl-Dickstein et al., 2015; Ho et al., 2020). In this objective, we explicitly train the energy landscape so gradient with respect to the energy function it can denoise and recover a motion plans $q_{1:T}$ across many differing levels of noise corruption $\{1, \ldots, S\}$ ranging from mostly correct motion paths to fully corrupted Gaussian noise trajectories. By shaping the gradient of the energy function to generate motion plans $q_{1:T}$ from arbitrary initialization trajectories, our learned energy landscape is able to effectively optimize for motion paths.

Formally, to train our potential, we use the energy based diffusion training objective in (Du et al., 2023) , where the gradient of energy function is trained to denoise noise corrupted motion plans $q_{1:T}^*$

$$\mathcal{L}_{\text{MSE}} = \|\epsilon - \nabla_{q_{1:T}} E_\theta(\sqrt{1 - \beta_s} q_{1:T}^i + \sqrt{\beta_s}\epsilon, s, q_{\text{start}}^i, q_{\text{end}}^i, C^i)\|^2 \tag{4}$$

---

**Algorithm 1** Code for Compositional Potential Based Planning

---

1: **Models:** compositional set of $N$ diffusion potential functions $E_\theta^i(q_{1:T}, t, q_{\text{start}}, q_{\text{end}}, C_i)$
2: **Hyperparameters:** horizon $T$, guidance scales $\omega_i$, denoising diffusion steps $S$
3: **Input:** start position $q_{\text{start}}$, goal position $q_{\text{goal}}$, $N$ constraints $C_{1:N}$
4: Initialize $q_{1:T}^S \sim \mathcal{N}(0, I)$
5: **for** $s = S \dots 1$ **do**
6:     # Combining Different Energy Potentials Together
7:     $\epsilon_{\text{comb}} = \nabla_{q_{1:T}} E_\theta(q_{1:T}^s, s, q_{\text{start}}, q_{\text{end}}, \varnothing) + \sum_{i=1}^N \omega_i \nabla_{q_{1:T}}(E_\theta^i(q_{1:T}^s, s, q_{\text{start}}, q_{\text{end}}, C_i) - E_\theta^i(q_{1:T}^s, s, q_{\text{start}}, q_{\text{end}}, \varnothing))$
8:     # Transit to Next Diffusion Time Step
9:     $q_{1:T}^{s-1} = q_{1:T}^s - \gamma\epsilon_{\text{comb}} + \xi, \quad \xi \sim \mathcal{N}(0, \sigma_s^2 I)$.
10: **end for**
11: **return**

---

where $\epsilon$ is sampled from Gaussian noise $\mathcal{N}(0, 1)$, $s \in \{1, 2, \dots S\}$ is the denoising diffusion step, and $\beta_s$ is the corresponding Gaussian noise corruption on a motion planning path $q_{1:T}^i$. We refer to $E_\theta$ as the *diffusion potential function*.

To optimize and sample from our diffusion potential function, we initialize a motion path $q_{1:T}^S$ at diffusion step $S$ from Gaussian noise $\mathcal{N}(0, 1)$ and optimize for motion path following the gradient of the energy function. We iteratively refine the motion $q_{1:T}^s$ across each diffusion step following

$$q_{1:T}^{s-1} = q_{1:T}^s - \gamma\epsilon_C + \xi, \quad \xi \sim \mathcal{N}(0, \sigma_s^2 I), \tag{5}$$

where $\epsilon_C = \epsilon_\varnothing - \omega(\nabla_{q_{1:T}} E_\theta(q_{1:T}, t, q_{\text{start}}, q_{\text{end}}, C) - \epsilon_\varnothing), \quad \epsilon_\varnothing = \nabla_{q_{1:T}} E_\theta(q_{1:T}, t, q_{\text{start}}, q_{\text{end}}, \varnothing)$ (6)

where $\gamma$ and $\sigma_s^2$ are diffusion specific scaling constants[1]. The final predicted motion path $q_{1:T}^*$ corresponds to the output $q_{1:T}^0$ after running $S$ steps of optimization from the diffusion potential function.

### 3.3 COMPOSING DIFFUSION POTENTIAL FUNCTIONS

Given two separate diffusion potential functions $E_\theta^1(\cdot)$ and $E_\theta^2(\cdot)$, encoding separate constraints in motion planning, we can likewise form a composite potential function $E_{\text{comb}}(\cdot) = E^1(\cdot) + E^2(\cdot)$ by directly summing the corresponding potentials. This potential function $E_{\text{comb}}$ will have low energy precisely at motion planning paths $q_{1:T}$ which satisfy both constraints, with sampling correspondings to optimizing this potential function.

To sample from the new diffusion potential function $E^{\text{comb}}$, we can follow

$$q_{1:T}^{t-1} = q_{1:T}^t - \gamma\nabla_{q_{1:T}}(E_\theta^{\text{comb}}(q_{1:T}, t, q_{\text{start}}, q_{\text{end}}, C)) + \xi, \quad \xi \sim \mathcal{N}(0, \sigma_t^2 I). \tag{7}$$

To further improve the composition, a more expensive MCMC procedure can be used to explicitly combine diffusion models (Du et al., 2023).

**Applications of Composing Potential Functions.** The ability to combine multiple separate potential functions for motion planning offers a variety of different ways to generalize and extend existing motion planning systems. First, in many motion planning problems, there are often a heterogenous set of different types of constraints or collisions that limit possible configuration space paths. For instance, in autonomous driving, constraints that can arise may include moving pedestrians, traffic lanes, road work or incoming cars. Oftentimes, we cannot enumerate all potential combinations, but we wish motion planning systems to be able to handle all possible combination of constraints. Jointly learning a single motion planning model for all constraints may be difficult, as at test time, we may see novel combinations that we do not have training data for. By learning separate diffusion potential fields for each constraint, we can combine them in an ad-hoc manner at test-time to deal with arbitrary sets of constraints. We provide two concrete implementations of composing potentials together as below and a detailed procedural in Algorithm 1.

**Generalization over More Obstacles** Suppose that the model is trained on environments with 4 obstacles, namely, $|C| = 4$. However, in the test time, we want to generalize to a more complex environment that has 6 obstacles $C' = \{o_1, o_2, o_3, o_4, o_5, o_6\}$. This can be achieved by adding the potentials evaluated on two sets of obstacles, where $C_1 = \{o_1, o_2, o_3, o_4\}$ and $C_2 = \{o_3, o_4, o_5, o_6\}$. This formulation can be further extended to $N$ sets of obstacles $C_{1:N}$ and the composite diffusion potential function is given by:

$$E_\theta^{\text{comb}}(q_{1:T}, t, q_{\text{start}}, q_{\text{end}}, C_{1:N}) = \sum_{i=1}^N E_\theta(q_{1:T}, t, q_{\text{start}}, q_{\text{end}}, C_i) \tag{8}$$

---

[1]A rescaling term at each diffusion step is omitted above for clarity

---

**Algorithm 2** Code for Refining Motion Plans

---

1: **Model:** compositional potential denoiser $f_\theta(q_{1:T}, t, q_{\text{start}}, q_{\text{end}}, C_{1:N})$
2: **Hyperparameters:** number of refine attempts $R$, noise scale $k$
3: **Input:** trajectory $q_{1:T}$, staet position $q_{\text{start}}$, goal position $q_{\text{goal}}$, $N$ constraints $C_{1:N}$
4: $S = \texttt{Get\_Collision\_Sections(q)}$ \qquad # A Set of Indices of Collision Sections in $q_{1:T}$
5: **for** $r = 1 \ldots R$ **do**
6: \qquad $q_{1:T}^k = \sqrt{\bar{\alpha}_k} q_{1:T} + (1 - \bar{\alpha}_k)\xi, \quad \xi \sim \mathcal{N}(0, \sigma_t^2 I)$ \qquad\qquad # Add Noise to $q_{1:T}$
7: \qquad $q' = f_\theta(q_{1:T}^k, k, q_{\text{start}}, q_{\text{end}}, C_{1:N})$, \qquad\qquad\qquad # Get new Denoised Trajectory
8: \qquad **for all** $s_i \in S$ **do**
9: \qquad\qquad **if** $\texttt{is\_section\_good}(q'[s_i])$ **then**
10: \qquad\qquad\qquad $q[s_i] = q'[s_i]; S = S \setminus s_i$ \qquad\qquad # Refine $q_{1:T}$ and Remove $s_i$ from set $S$
11: \qquad\qquad **end if**
12: \qquad **end for**
13: **end for**
14: **return** $q$

---

|  |  |  |  |
| is success: ■ len:48 | is success: ■ len:48 | is success: ■ len:48 | is success: ■ len:48 |
| Initial Noise | Proposal Plan | Partially Noisy | Replanned |

Figure 2: **Visualization of the Motion Refining Scheme**. A proposal plan is first generated by denoising an initial Gaussian noise. If collision is detected, a small noise is first added to the proposal and the new plan is generated based on the partially noisy trajectory.

**Generalization over Static and Dynamic Obstacles.** Many real-life scenarios involves *dynamic* real-time interaction. For instance, to construct motion plan for an autonomous vehicle, we must both avoid static lane obstacles as well as dynamically moving cars. While static obstacles are often known a priori, the motion patterns of dynamics obstacles often change with time, making it advantageous to be able to combine different dynamic constraints with static ones. We can directly implement this by using a diffusion potential function $E_{\theta_s}^i$ that only trained on static obstacles $C_i^s$ and a diffusion potential function $E_{\theta_d}^j$ that only trained on dynamic obstacles $C_j^d$, we can obtain the static&dynamic potential by adding $E_{\theta_s}^i$ and $E_{\theta_d}^j$. In a more general form, to condition on a set of $N_1$ static obstacles $C_{1:N_1}^s$ with their potential diffusion functions $E_{\theta_s}^{1:N_1}$ and a set of $N_2$ dynamic $C_{1:N_2}^d$ obstacles with their potential diffusion functions $E_{\theta_d}^{1:N_2}$, the composite diffusion potential function is then written as:

$$E_\theta^{\text{comb}}(q_{1:T}, t, q_{\text{start}}, q_{\text{end}}, [C_{1:N_1}^s, C_{1:N_2}^d]) = \sum_{i=1}^{N_1} E_{\theta_s}^i(q_{1:T}, t, q_{\text{start}}, q_{\text{end}}, C_i^s) + \sum_{j=1}^{N_2} E_{\theta_d}^j(q_{1:T}, t, q_{\text{start}}, q_{\text{end}}, C_j^d) \quad (9)$$

## 3.4 REFINING MOTION PLANS

In practice, the predicted motion plan $q_{1:T}$ might occasionally contains sections that violate the constraints of the environment (i.e., collide with obstacles). To solve this issue, both classical and learned motion planners (Kuffner & LaValle, 2000; Qureshi et al., 2019) provide mechanisms to refine trajectories subject to collisions in configuration space.

With diffusion potential fields, we can likewise refine a trajectory, $q_{1:T}$ with collision, by locally perturbing it into a noisy trajectory $q_{1:T}^k$ defined by the $k$th step of the diffusion forward process:

$$q_{1:T}^k = \sqrt{\bar{\alpha}_k} q_{1:T} + (1 - \bar{\alpha}_k)\xi, \quad \xi \sim \mathcal{N}(0, \sigma_t^2 I). \quad (10)$$

A new motion plan $q'_{1:T}$ can be obtained by denoising the noisy trajectory following Equation 5. To be simple, let

$$q'_{1:T} = f_\theta(q_{1:T^k}, k, q_{\text{start}}, q_{\text{end}}, C_{1:N}) \quad (11)$$

where $f_\theta(.)$ is a iterative diffusion potential denoiser that output the clean trajectory. The warm-start denoising scheme enables faster planning and is more efficient, especially important for those energy-critical mobile agents. We will then replace the collision section in $q_{1:T}$ with corresponding section in $q'_{1:T}$ when the new section is collision-free. This refining procedural can be repeated

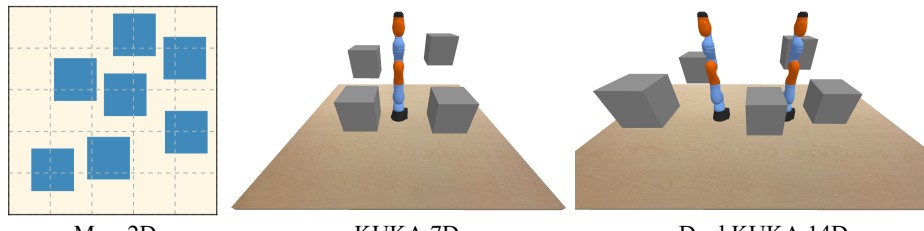

Maze2D                    KUKA 7D                    Dual KUKA 14D

Figure 3: **Environment Demonstration.** a) Maze2D: a point robot moving in 2D workspace with the high-lighted block as obstacles. b) KUKA: robot manipulator with 7 DoF operating on a tabletop. The grey cuboids are obstacles. c) Dual KUKA14D: Two side by side KUKA manipulators operate simultaneously, where the dimension of the configuration space is 14.

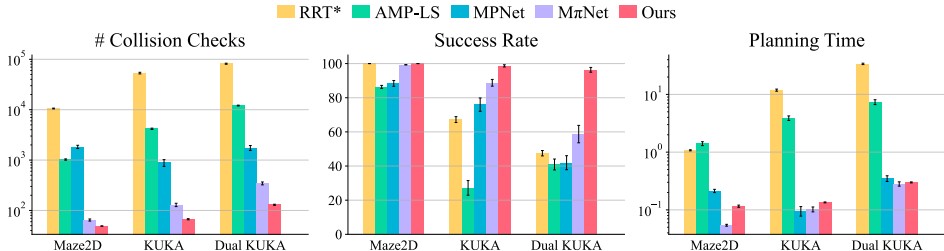

Figure 4: **Quantitative Comparisons in Motion Planning Environments.** Our method outperforms the sampling-based planner and all other learning-based motion planning approaches on all metrics across a set of different environments. From left to right: a) number of collision checks, b) success rate, c) planning time.

until a desired trajectory is found. Algorithm 2 displays the complete refining pipeline and Figure 2 provides a corresponding visualization.

## 4 EXPERIMENTS

In this section, we firstly describe our environments and baselines in Section 4.1. Next, in Section 4.2, we discuss our experiments on base environments and motion refining algorithm. Following, in Section 4.3, we present the compositionality results by evaluating our motion planner on composite environments. Then, we describe the real world motion planning performance in Section 4.4.

### 4.1 ENVIRONMENTS AND BASELINES

We first classify the environments that we evaluated on to 4 categories by the level of generalization capability:

- **Base Environments:** same number of constraints as in training; constraints sampled from the same distribution;

- **Composite Same Environment:** more constraints than training phase, constraints sampled from the same distribution;

- **Composite Different Environment:** more constraints than training phase, constraints sampled from different distributions.

- **Real World Motion Planning Environments.**

Concretely, we propose three simulated motion planning environments with increasing difficulty as shown in Figure 3:

**Maze2D**   A point-robot moving in a 2D workspace. The configuration space is the x-y coordinate of the robot. The task is to generate a 2D trajectory navigate through the workspace without any collision with obstacles. We offer two variants: *Static Maze2D* where obstacles stay in the same locations and *Dynamic Maze2D* where obstacles are moving in randomly generated linear trajectories.

**Kuka7D**   A KUKA arm of 7 DoF operating on a tabletop. Obstacles are randomly placed in the 3D workspace. The start and goal are given as the 7 joint states of the KUKA arm.

**Dual KUKA**   Two KUKA arms are placed side by side on a tabletop and operate simultaneously with a total configuration space of 14 DoF. A successful trajectory should have both arms arrived in their goal states and should not have any self-collision or collision with obstacles.

**Baselines**   We compare our methods with the classic sampling-based planning baselines RRT* (Karaman & Frazzoli, 2011), P-RRT* (Qureshi & Ayaz, 2016), BIT* (Gammell et al., 2015)

| Env | $R = 3$ Before | $R = 3$ After | $R = 5$ Before | $R = 5$ After | $R = 10$ Before | $R = 10$ After |
|---|---|---|---|---|---|---|
| Maze2D | 96.25 | 99.75 | 95.25 | 99.00 | 95.75 | 100.00 |
| KUKA | 71.25 | 90.00 | 69.50 | 94.30 | 69.75 | 94.75 |
| Dual KUKA | 45.50 | 69.75 | 47.25 | 77.25 | 47.00 | 80.75 |

Table 1: **Quantitative Results of Refining Motion Plans**. Success rate before and after motion refining. $R$ denotes the number of refine attempts. The proposed method consistently boost success rate on three base environments.

| Method | Static 1 + Static 2 Success | Time | Check |
|---|---|---|---|
| RRT* | 99.90 | 2.15 | 19k+ |
| Ours | **100.00** | **0.38** | **71.86** |

Table 2: **Quantitative Results on Composite Different Environments.** Two static Maze2D with different types of obstacles are combined at test time.

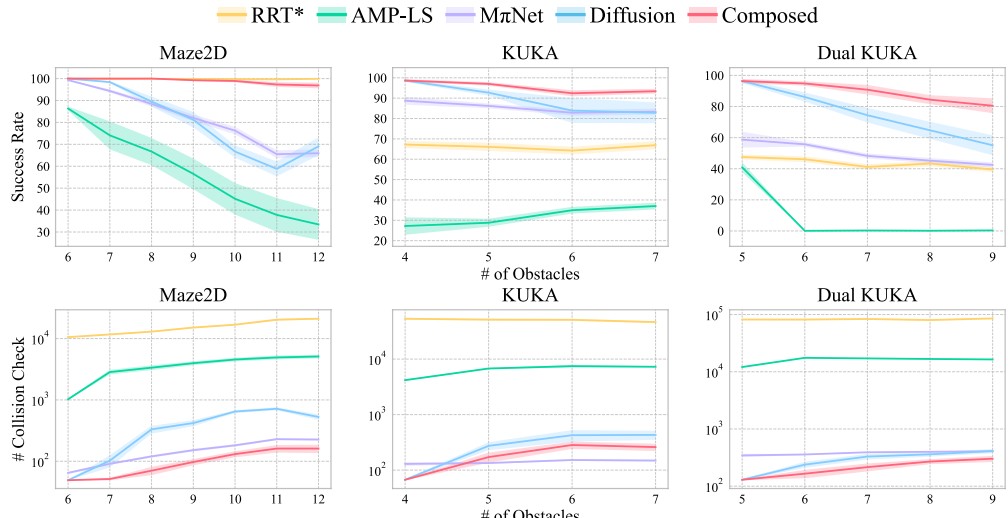

Figure 5: **Compositional Generalization.** Quantitative comparisons of different planner on compositional environment. The shaded area indicates the standard error across the mean of all tested environments. The leftmost column reports the results on the same number of obstacles that the models trained on. We report The composite model outperforms all other baseline by a margin, only except that in Maze2D, where RRT* is on par with our model, but with order of magnitude of more collision checks.

and SIPP (Phillips & Likhachev, 2011), traditional potential-based method RMP (Ratliff et al., 2018), and several learning-based motion planners: MPNet (Qureshi et al., 2019), MπNet (Fishman et al., 2023), and AMP-LS (Yamada et al., 2023). MPNet is trained on trajectories with sparse waypoints and use MLPs to encode environment configuration and predict the next position. In contrast, MπNet is trained on dense trajectory waypoints and predicts the movement vector instead of directly the next position. AMP-LS encodes the robot pose into a latent feature and approaching the goal pose by using the gradient of hand-crafted losses to update the latent. A sequence of latents are then decoded and form a trajectory. In evaluation, all start/goal poses and environment configurations are unseen to the model. For each experiment, we evaluate on 100 different environments with 20 problems each.

### 4.2 MOTION PLANNING PERFORMANCE ON BASE ENVIRONMENTS

We first evaluate our method on motion planning in each base environments: randomly generated environments that follow the same procedural generation pipeline as the training environments. Qualitative results are shown in Figure 4 and Table VIII. We include the full details of evaluation setup in Section A.2.3.

**Comparison to Sampling-based Planner**    We compare our method to traditional sampling-based RRT*(Karaman & Frazzoli, 2011). The success rate of RRT* suffers from a significant degradation when the dimension of the configuration space increases. In addition, the planning time of the sampling-based planner rises dramatically as the dimension of the problems increases. However, the planning time of our method performs steadily across all environments, namely, 0.116s, 0.135s, 0.299s and with order of magnitude less collision check.

**Comparison to Learning-based Planners**    We also compare to three other learning-based motion planning baselines: MPNet, MπNet, and AMP-LS, as displayed in Figure 4 and 6. We can see that our method outperform all the learning baseline in both success rate and number of collision check.

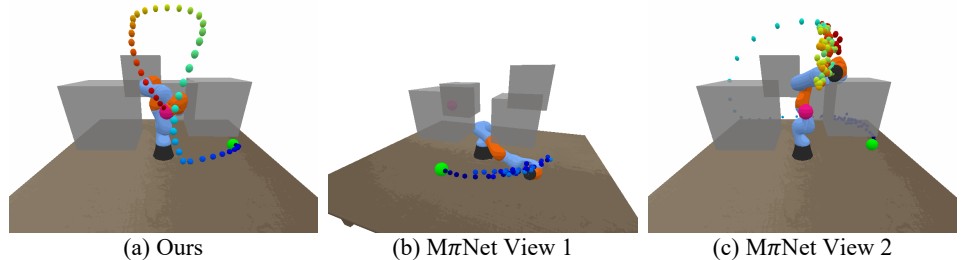

|  (a) Ours  |  (b) MπNet View 1  |  (c) MπNet View 2  |

Figure 6: **Qualitative Motion Plan in KUKA Environment.** Obstacles are shown in transparent grey for clearer view. Our method, in column (a), generates an end-to-end, smooth trajectory. In column (b) and (c) show the trajectory generated by MπNet from two different viewing angles. The proposed trajectory traverses from the other direction that requires more movement, is frequently stuck in local geometry, and finally fails to reach the goal state.

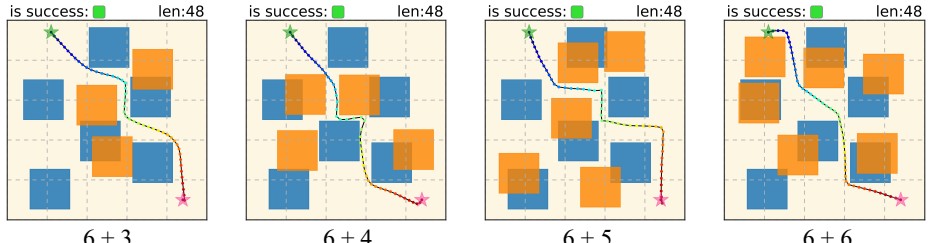

|  6 + 3  |  6 + 4  |  6 + 5  |  6 + 6  |

Figure 7: **Qualitative Compositionality Generalization over More Obstacles.** Two models that trained on only six obstacles are composed and tested on out-of-distribution environments, with 9, 10, 11, 12 obstacles, respectively.

| Method | Base Dynamic | | | Static 1 + Dynamic | | | Static 2 + Dynamic | | |
| --- | --- | --- | --- | --- | --- | --- | --- | --- | --- |
|  | Success | Time | Check | Success | Time | Check | Success | Time | Check |
| SIPP | 69.85 | 32.21 | 1M+ | 70.40 | 185.50 | 1.7M+ | 73.95 | 98.66 | 1.3M+ |
| Ours | **99.65** | **0.12** | **49.26** | **97.35** | **3.72** | 213.97 | 97.95 | 3.63 | 177.31 |

Table 3: **Quantitative Results on Base Dynamic and Static + Dynamic on Maze2D.** Static 1 and Static 2 refer to two different static Maze2D environments. Our method outperforms the sampling-based planner by a large margin.

Notably, in Dual KUKA, our method led the the state-of-the-art learning-based planner MπNet by 37% while with 3 times less of collision checks. We also observe that the planning time of it is slightly shorter than ours, even though it requires a higher number of collision checks. Note that the gap is closing as the dimension of the environment increases – in practice in the real world, we believe this gap will be further eliminated where collision checks is much more expensive.

**Motion Refining**    We present quantitative and qualitative results of refining motion plans, as shown in Table 1 and Figure 2. The gain of refining motion plans increases as the dimensionality of the environment increases. As in Table 1, the success rate generally increases as we increase the number of refining attempts $R$, but the gain gradually saturates in 10 attempts. In this case, the proposed trajectory probably suffers from a catastrophic collision and the model might need to resample a trajectory from a pure noise.

## 4.3 COMPOSITIONALITY

**Composing Obstacles**    We first evaluate the compositionality by adding obstacles to the environments. A qualitative visualization of a composite Maze2D environment is given in Figure 7, where we train our model on 6 obstacles and evaluate on environments with up to 12 obstacles. The blue blocks indicate 6 obstacles as in training distribution, while the orange blocks indicate out-of-distribution additional obstacles. As we can see, the composed model effectively proposes different trajectories according to the presented obstacles by sampling poses from the region with low composite potential. We report the full quantitative results in Figure 5 and Table XI.

**Composing Multiple Constraints**    We then investigate the compositionality to combine two different diffusion potential functions together, (i.e., models trained on completely different environments ). Specifically, we first train a model on 6 small obstacles and a model on 3 large obstacles and evaluate on environments where both the small and large obstacles are presented. The qualitative results is shown in Table 2. Moreover, we want to compose the two aforementioned models trained

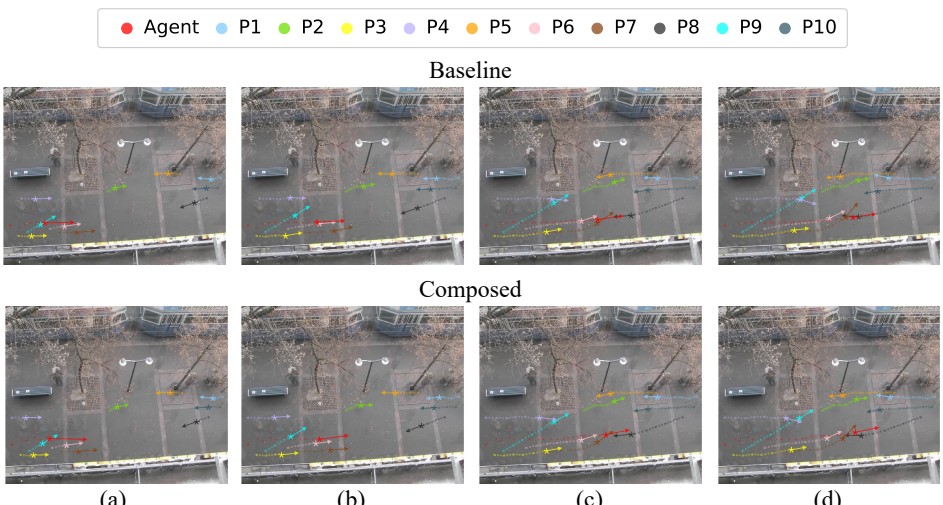

Figure 8: **Qualitative Real World Motion Plans, *Hotel* Scene.** The composed model provides long-horizon motion plan that avoid 10 pedestrians, while only trained on 5 pedestrians. In column (a) and (b), the composed plan is aware of P1 (cyan) and P6 (pink) and overtakes them from above, while the baseline model runs into them. In column (c), the composed motion plan chooses to move faster so as to pass through the intersection with P7 (brown) before P7 arrives, but the baseline motion plan results in a collision due to its slower speed. In column (d), the composed plan choose to go upward to avoid the oncoming P8 (black).

on static environments with another model that trained on dynamic environments. Hence, we test the composed model on environments where both static and dynamic obstacles are presented. We named the environments static 1 + dynamic and static 2 + dynamic, respectively. The quantitative results of the base dynamic environment and static + dynamic environments are shown in Table 3 and the qualitative results are in Figure X.

## 4.4 REAL WORLD

Finally, we evaluate the effectiveness of our method on the real world ETH\UCY(Pellegrini et al., 2010; Lerner et al., 2007) dataset. The dataset group we used consists of 5 scenes (*ETH, Hotel, Zara01, Zara02, UNIV*), where each scene contains human trajectories in world-coordinates collected by manual annotation from bird-eye-view camera. Our focus is to investigate if our model can propose successful trajectories given the start and goal locations of an agent in a random, cluttered street-level real-world interaction. Specifically, the planner is trained to predict the trajectory of the agent (highlighted in red), conditioned on the trajectories of 5 other pedestrians. Data from all the scenes are used when training and evaluate on unseen combination of start, goal, and surrounding pedestrian trajectories. In Figure XI, we present the qualitative results where 5 other pedestrians are presented. We also evaluate on 10 presented pedestrians by composing the two potential functions constrained by 5 pedestrians each, as illustrated in Figure 8.

## 5 DISCUSSION

**Limitations.** Our existing formulation of potential based diffusion motion planner has several limitations. First, although our motion trajectory is accurate, it is often suboptimal, e.g., there exists a shorter path from start to goal. This may be addressed by adding an additional potential to reach the goal as soon as possible. Second, our approach to composing potentials scales linearly with the number of composed models, requiring significantly more computation power with additional models. This can remedied by having different potential operate on shared features in a network.

**Conclusion.** In this work, we have introduced the potential based diffusion motion planner. We first formulate our potential diffusion motion planner and describe its connections and advantages over traditional potential based planner. We illustrate the motion planning performance of our approach in terms of success rate, planning time, and the number of collision checks over motion planning problems with dimensionality of 2D, 7D, 14D. We further illustrate the compositionality of apporach, enabling generalization to both new object and new combinations of motion constraints. Finally, we illustrate the potential of our work on real world scenes with multi-agent interaction.

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

# A  APPENDIX

In this appendix, we first present our dataset details in Section A.1. Next, we provide implementation details in Section A.2, including model architecture and training and evaluation hyperparameters. Section A.3 provides additional quantitative and qualitative results.

## A.1  DATASET DETAILS

In this section, we present details of the three base environments. Each dataset is consists of feasible trajectories of randomly sampled start and goal position. All datasets are collected by BIT*(Gammell et al., 2015). All environments except Maze2D are simulated via PyBullet (Coumans & Bai, 2016–2021).

**Maze2D**  The workspace is a $5 \times 5$ square where the obstacle is square of size $1 \times 1$. For simplicity, the volume of the robot agent is ignored – collision happens when the location is inside the region of an obstacle. The training data contains 3,000 different environment configurations and 25,000 waypoints for each environments.

**KUKA**  The KUKA arm is placed at the center $(0, 0, 0)$ of world coordinate and obstacles are given as cubic of length 0.4 meter and are randomly placed in the surrounding of the robot. The training data contains 2,000 different environment configurations and 25,000 waypoints for each environments.

**Dual KUKA**  For our Dual KUKA environment, one KUKA arm is placed at world coordinate $(-0.5, 0, 0)$ and the other at $(0.5, 0, 0)$. The training data contains 2,500 different environment configurations and 25,000 waypoints for each environments.

## A.2  IMPLEMENTATION DETAILS

**Software:**  The computation platform is installed with Red Hat 7.9, Python 3.8, PyTorch 1.10.1, and Cuda 11.1

**Hardware:**  For each of our experiments, we used 1 RTX 3090 GPU.

### A.2.1  ENERGY-BASED DIFFUSION MODEL

**Model Architecture**  We represent the diffusion potential model $f_\theta$ consisting of a CNN trajectory denoiser based on U-Net similar to (Ajay et al., 2023) and a constraint (i.e., environment configuration) encoder. The U-Net contains repeated residual blocks where each block consisted of two temporal convolutions followed by GroupNorm and SiLU nonlinearity (Hendrycks & Gimpel, 2016). The constraint encoder use the Transformer encoder structure (Vaswani et al., 2017), whose input is as a set of obstacle locations or obstacle trajectories. We remove the positional embedding, since the obstacles information should be permutation invariance with each other. We concatenate the learned class token from transformer with the time embedding and feed the concatenated tensor to temporal convolution blocks in U-Net for denoising. More details of the models are shown in Table A.2.1 and Table A.2.1. Note that we do not further explore the selection of the concrete model architecture, but we believe that some more advanced architectures could further improve our performance.

| Hyperparameters | Value |
|---|---|
| Base Feature Channels | 64 |
| Feature Dimension Scale | (64, 256, 512) |
| Groups in GroupNorm | 8 |
| Nonlinearity | SiLU |

Table IV: **Hyperparameter of U-Net.**

| Hyperparameters | Value |
|---|---|
| Base Embedding Channel | 64 |
| Transformer Layers | 3 |
| Attention Heads | 1 |
| Nonlinearity | SiLU |

Table V: **Hyperparameter of Constraint Encoder.**

**Energy Parameterization**  We use L2 energy-parameterization as given in equation 12. For more details on Energy-based Diffusion Model, please refer to (Du et al., 2023).

$$E_\theta^{L2}(x, t) = \frac{1}{2}||f_\theta(x, t)||^2 \tag{12}$$

### A.2.2  TRAINING DETAILS

**Training Pipeline**  In training, the dataloader randomly samples trajectories of length equal to the training horizon from the whole dataset. We provide a detailed hyperparamter of training our model

in Table VI. We do not apply any hyperparameter search nor learning rate scheduler. We typically train our model for two days, but we observe the performance are close to saturated within one day.

| Hyperparameters | Value |
|---|---|
| Horizon | 48 |
| Diffusion Time Step | 100 |
| Probability of Condition Dropout | 0.2 |
| Iterations | 2M |
| Batch Size | 512 |
| Optimizer | Adam |
| Learning Rate | 2e-4 |

Table VI: **Hyperparameters: Training on Maze2D Environment**

### A.2.3 EVALUATION DETAILS

**Our Evaluation Pipeline**  The input of the planner is the start state, goal state and environment configuration, and the outputs are the proposed trajectories. In test time, we only sample 10 timesteps by using DDIM (Song et al., 2020) with intermediate noise scale eta set to 0. The evaluation pipeline of our model consists of three phases: Propose Motion Plan Candidates, Candidate Selection, and Motion Refining. The planner first generates multiple candidate trajectories as given in Algorithm 1. It then accesses to the environment configuration to select a successful trajectory from the candidates. Finally, if no desired candidate is found, it will execute the motion refining as in Algorithm 2. Hyperparameter used in evaluation is detailed in Table VII.

**Baselines Evaluation Pipeline**  We try our best to re-implement every baseline and follow their original setting. For MPNet (Qureshi et al., 2019), we follow their implementation and use bidirectional path generation in test time. As for AMP-LS, we implement a Variional Auto-Encoder (VAE) (Kingma & Welling, 2013) to encode the robot pose state and leverage GECO loss (Rezende & Viola, 2018) in path optimization. M$\pi$Net does not provide a replan scheme in their design. For fair comparison, we boost M$\pi$Net with replan by backtracing to previous timestep and adding random noise for restart when collision is detected.

| Hyperparameters | Value |
|---|---|
| Horizon | 48 |
| DDIM Time Step | 10 |
| DDIM eta | 0.0 |
| Guidance Scale | 2.0 |
| # of Trajectory Candidate | 10 |
| # of Refine Attempts $R$ | 5 |
| Refine Noise Scale $k$ | 3 |

Table VII: **Hyperparameters: Evaluation on Maze2D Environment**

### A.3 ADDITIONAL RESULTS

In this section, we provide more quantitative results in A.3.1 and extra qualitative results in A.3.2.

### A.3.1 QUANTITATIVE RESULTS

**Performance on Base Environment**  We provide detailed numerical motion planning results on three base environments in Table VIII. The corresponding visualization is given in Figure 4. We report three main motion planning metrics: Success rate, Planning time and Number of collision checks. Our method consistently outperforms all other baselines.

**Performance on Composite Same Environment**  We provide quantitative results on out-of-distribution environments by adding more obstacles than training time. We further provide a baseline *Diffusion* in which we do not compose potentials. Our composed model demonstrate superior efficiency over all benchmarks. Note that RRT* generally requires two order of magnitude of more collision checks than our method for planning one path.

| Method | Maze2D | | | KUKA | | | Dual KUKA | | |
|--------|---------|------|-------|---------|------|----------|-----------|-------|-----------|
| | Success | Time | Check | Success | Time | Check | Success | Time | Check |
| RRT* | 100.00 | 1.08 | 10561.73 | 67.20 | 11.86 | 53282.03 | 47.50 | 33.80 | 81759.17 |
| P-RRT* | 99.95 | 1.34 | 15697.09 | 66.20 | 12.72 | 53590.14 | 47.40 | 33.98 | 104820.10 |
| BIT* | 100.00 | 0.21 | 1894.20 | **99.20** | 1.17 | 7988.53 | 94.95 | 4.60 | 20689.98 |
| AMP-LS | 86.35 | 1.41 | 1025.36 | 27.20 | 3.89 | 4176.40 | 40.90 | 7.38 | 12091.33 |
| MPNet | 88.40 | 0.21 | 1830.90 | 75.95 | 0.10 | 885.41 | 41.95 | 0.35 | 1750.17 |
| M$\pi$Net | 99.30 | 0.05 | 64.66 | 88.75 | 0.10 | 129.15 | 58.70 | 0.28 | 345.03 |
| Ours | **100.00** | **0.12** | **49.07** | 98.65 | 0.13 | 67.02 | **96.35** | 0.30 | **129.63** |

Table VIII: **Quantitative Motion Planning Performance.** Detailed numerical results corresponding to Figure 4. Our method is able to efficiently propose paths and outperforms other baseline planners, especially in the hardest Dual KUKA environment.

**Performance on ETH/UCY Dataset**   Current robotic path planning problem is usually limited to simulation environments and there is no widely-used real-world benchmark. Since pedestrian trajectory is naturally a kind of demonstrations of human planning, we leverage the ETH/UCY Dataset to evaluate real-world motion planning ability. Specifically, the entire dataset is comprised of 6 scenes: *ETH, Hotel, Zara01, Zara02, Students01, Students03*, where the models are trained on 5 scenes and tested on the held-out scene. Similar to our simulation environments, the model takes as input the position of start, goal, and other pedestrians, and outputs a predicted motion plan. The predicted motion plan is desired to be as close as possible to the real human trajectory and thus we report the Average Displacement Error (ADE) as defined in equation 13:

$$\text{ADE} = \frac{\sum\limits_{i \in N} \sum\limits_{t \in T} ||q_t^i - \hat{q}_t^i||_2}{N \times T} \tag{13}$$

where $\hat{q}_t^i \in \mathbb{R}^2$ is the step $t$ in the $i$th predicted path and $q_t^i$ is the corresponding ground truth. ADE measure the similarity between the predicted trajectories and human trajectories. A smaller ADE value indicates the predictions are closer to the real human behavior.

| Method | *ETH* | | *Hotel* | | *Zara01* | | *Zara02* | | *Students01* | | *Students03* | |
|--------|-------|------|---------|------|----------|------|----------|------|--------------|------|--------------|------|
| | ADE↓ | Time | ADE | Time | ADE | Time | ADE | Time | ADE | Time | ADE | Time |
| MPNet | 18.11 | 0.12 | 28.60 | 0.11 | 11.06 | 0.12 | 17.22 | 0.11 | 10.37 | 0.12 | 8.93 | 0.11 |
| M$\pi$Net | 37.70 | 0.26 | 44.49 | 0.29 | 1.14 | 0.22 | 13.66 | 0.23 | 12.76 | 0.18 | 1.54 | 0.22 |
| Ours | **0.94** | **0.17** | **5.20** | **0.17** | **0.35** | **0.17** | **0.38** | **0.17** | **0.52** | **0.17** | **0.89** | **0.17** |

Table IX: **Quantitative Results on real-world ETH/UCY Dataset.** We adopt the ADE metric to show the similarity between the predicted motion plans and real human motion trajectories on unseen scenarios. All motion plans are in the world coordinate as given in the dataset. Our method can precisely mimic human motion behaviors in most scenes, while both MPNet and M$\pi$Net cause drastic error compared to real human trajectories.

The per-scene quantitative performance is shown in Table IX. Each scene is captured at different time of a day or different location, resulting in different data distribution. The ADE is relatively smaller in *Zara* and *Students* because the model can see a similar counterpart scene at training, e.g., train set includes *Zara01* when tested on *Zara02*. By contrast, *ETH* or *Hotel* are more unique to other scenes (e.g., contain different scene layout and human behavior patterns) and thus causing higher evaluation error. As shown in Table IX, MPNet consistently falls short of the ADE, though with slightly faster speed. We observe that M$\pi$Net can produce reasonable motion plans in many cases, but it occasionally predicts random values, which causes significant deviation from the target and leads to the notbaly larger ADE. Our method demonstrates better generalizability and stability by precisely mimicking the human trajectories in most held-out scenes. We also notice that in *Hotel*, all methods suffer from a severe surge in ADE. Our speculation is that the *Hotel* has different walkable world-coordination and in addition, it contains various unseen types of pedestrians motion pattern, such as slowly pacing people that are chatting or waiting, people stepping on or off the train.

**Performance on Concave Obstacles**   We create a Maze2D environment with 7 concave obstacles as shown in Figure IX to further demonstrate the capability of our learned potential method to avoid local minima. As shown in Table X, all methods subject to a certain decline in performance in the more difficult concave environments. Notably, our potential-based diffusion motion planner can still

solve all the problems, on par with the sampling-based planners, while traditional potential-based planner, RMP, exhibits a significant decline in success rate.

| | Maze2D – Convex | | Maze2D – Concave | |
|---|---|---|---|---|
| Method | Success | Time | Success | Time |
| RRT* | 100.0 | 1.08 | 100.0 | 2.53 |
| P-RRT* | 99.95 | 1.34 | 99.9 | 3.31 |
| BIT* | 100.0 | 0.21 | 100.0 | 0.45 |
| MPNet | 88.4 | 0.21 | 84.3 | 0.38 |
| MπNet | 99.3 | **0.05** | 98.7 | **0.06** |
| RMP | 64.9 | 0.13 | 28.0 | 0.34 |
| Ours | **100.0** | **0.12** | **100.0** | **0.15** |

Table X: **Quantitative Performance with Convex and Concave Obstacles.** Motion planning performance on the Maze2D environments with 6 convex obstacles and 7 concave obstacles. Obstacles are randomly placed and example visualizations are shown in Figure 3 and Figure IX, respectively. Reported results are averaged across 100 different environment configurations with 20 problems in each configuration. Both our method, RRT*, and BIT* can successfully solve all the motion planning problems, while our method requires less planning time.

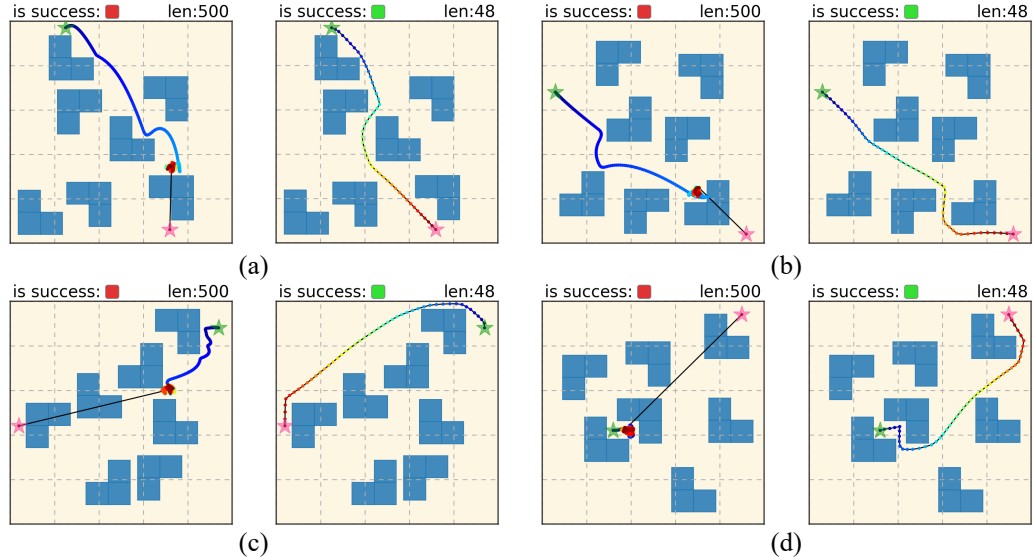

Figure IX: **Qualitative Performance on Environments with Concave Obstacles.** Trajectories generated by RMP and our method on 4 motion planning problems are shown (left: RMP; right: ours). Traditional potential-based method tends to stuck in the local minima, while our method can generate smooth and low-cost trajectories without collision. The green star indicates the star pose and the red star indicates the goal pose.

### A.3.2 QUALITATIVE RESULTS

**Static 2 + Dynamic Environment** In Figure X, we show our results on a Composite Different Environment as described in Section 4.1. Specifically, one component environment consists of a large blue block moving along the grey trajectories (i.e. *dynamic*) and the other component environment consists of three orange *static* obstacles.

**Base Real World Motion Planning** We visualize the motion trajectory planned by our model in *Zara02* scene where five other pedestrians are presented in Figure XI. In the given scenario, our agent (highlighted in red) enters the scene alongside with the P2 (yellow) and enter an intersection with four oncoming pedestrians. The trajectory planned by our model first chooses to follow the P2 (shown in T = 10) and then crosses the intersection without interrupting any other pedestrians (shown in T = 22).

### A.4 PROOF OF OPTIMALITY AND COMPLETENESS

In this section, we will show the probabilistic completeness and optimality of our method.

|  | Method | 6 + 2 | | 6 + 4 | | 6 + 6 | |
|  |  | Success | Check | Success | Check | Success | Check |
|---|---|---|---|---|---|---|---|
| Maze2D | RRT* | 99.90 | 12986.05 | **99.70** | 16842.79 | **99.90** | 21089.77 |
|  | AMP-LS | 66.70 | 3357.64 | 45.15 | 4560.38 | 33.45 | 5107.20 |
|  | MπNet | 88.45 | 120.00 | 76.25 | 181.77 | 65.95 | 226.01 |
|  | Diffusion | 89.50 | 331.30 | 66.70 | 646.80 | 69.10 | 527.75 |
|  | Composed | **100.00** | **70.21** | 98.90 | **130.78** | 96.85 | **161.03** |

|  | Method | 4 + 1 | | 4 + 2 | | 4 + 3 | |
|  |  | Success | Check | Success | Check | Success | Check |
|---|---|---|---|---|---|---|---|
| KUKA | RRT* | 66.10 | 51405.21 | 64.25 | 50993.57 | 66.90 | 46399.91 |
|  | AMP-LS | 28.80 | 6767.00 | 34.95 | 7461.55 | 37.00 | 7277.37 |
|  | MπNet | 86.20 | **134.13** | 82.85 | **152.03** | 83.35 | **148.67** |
|  | Diffusion | 92.70 | 273.29 | 83.90 | 425.54 | 82.80 | 430.78 |
|  | Composed | **97.00** | 171.54 | **92.40** | 283.35 | **93.40** | 259.90 |

|  | Method | 5 + 1 | | 5 + 2 | | 5 + 3 | |
|  |  | Success | Check | Success | Check | Success | Check |
|---|---|---|---|---|---|---|---|
| Dual KUKA | RRT* | 46.10 | 81670.24 | 41.15 | 83559.30 | 43.35 | 80090.60 |
|  | AMP-LS | 0.10 | 17512.00 | 0.30 | 17107.36 | 0.15 | 16725.14 |
|  | MπNet | 55.70 | 358.28 | 48.25 | 391.52 | 45.20 | 399.11 |
|  | Diffusion | 86.10 | 237.93 | 74.40 | 329.72 | 64.70 | 360.91 |
|  | Composed | **94.80** | **165.42** | **90.80** | **215.30** | **84.40** | **269.73** |

Table XI: **Compositional Generalization over Increased Obstacles.** Detailed results corresponding to Figure 5. In the top row, the *left digit* indicates the number of obstacles that the model trained on; the *right digit* represents the number of additional obstacles. Compared to other learning-based planner, our method smoothly generalizes to out-of-distribution test-time environments by compositionality while others suffer from a noticeable decrease in success rate.

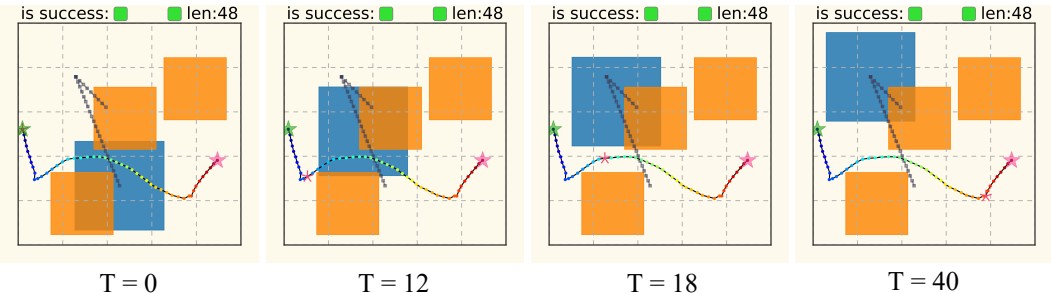

|  |  |  |  |
|---|---|---|---|
| T = 0 | T = 12 | T = 18 | T = 40 |

Figure X: **Qualitative Compositionality Generalization over Static + Dynamic Obstacles.** Zoom in for clearer view. The current position of the agent is shown in the pink asterisk. The planned trajectory first goes down in order to wait for the large moving obstacles to pass through and then goes left toward the goal, while still being aware of other orange static obstacles in the environment.

### A.4.1 PROBABILISTIC COMPLETENESS

Let $f_\theta(q_{1:T})$ denote the probability density function of the output distribution $\mathcal{D}_o$ of our diffusion model. In such learned neural distribution, all data points $q_{1:T}$ are assigned positive density, that is,

$$\forall q_{1:T}, \ f_\theta(q_{1:T}) > 0 \tag{14}$$

Define $\mathcal{J}_c$ as the a set of all valid trajectories subject to constraint $C$. There exists a small interval in the vicinity of a random trajectory $q_{1:T}^c \in \mathcal{J}_c$, such that

$$[q_{1:T}^c - \tau, q_{1:T}^c + \tau] \subseteq \mathcal{J}_c, \quad \tau > 0, \tag{15}$$

i.e., all trajectories in the interval satisfy the given constraint $C$. Let $\mathbb{P}_\tau$ denote the probability for our model to sample a trajectory from the interval, and according to equation 14 we have,

$$\mathbb{P}_\tau = \int_{q_{1:T}^c - \tau}^{q_{1:T}^c + \tau} f(x)dx > 0 \tag{16}$$

Let $A_n$ denote the event that there is at least one trajectory $q_{1:T} \in \mathcal{J}_c$ among $n$ sampled trajectories. Clearly, as the number of samples approaches infinity, event $A$ will happen almost surely, i.e.,

$$\lim_{n \to \infty} \mathbb{P}(A_n) = 1 \tag{17}$$

Hence, our method is probabilistically complete.

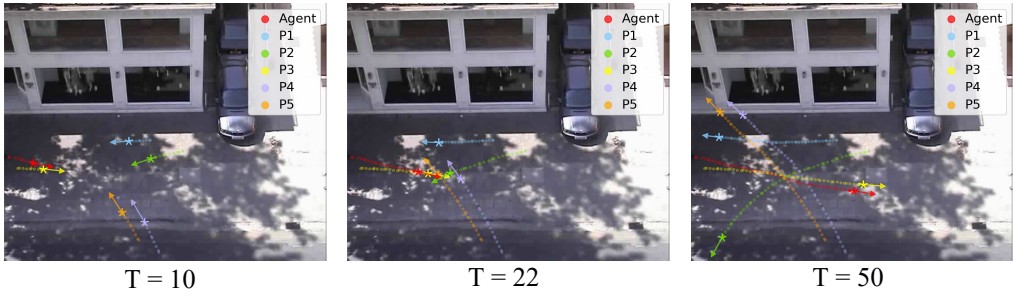

| T = 10 | T = 22 | T = 50 |

Figure XI: **Qualitative Real World Motion Plan, *Zara02* Scene.** Red indicates the trajectory planned by our model, while other colors represents 5 unseen pedestrians in the surrounding. Our motion plan passes through the intersection, without any collision or discontinuity (an abrupt stop).

### A.4.2 OPTIMALITY

Theoretically, with sufficient training data and according to the *Universal Approximation Theorem*, the optimal energy function $E_\theta^*(.)$ can be learned by our model. Hence, let $f_\theta^*(q_{1:T}, C)$ denote the probability density function of the optimal output distribution subject to the constraint $C$ and $q_{1:T}^*$ denote the corresponding optimal trajectory. Then,

$$\forall C, \quad f_\theta^*(q_{1:T}^*) \to \infty. \tag{18}$$

Thus, assume that an optimal model is learned, our model can generate optimal trajectories almost surely.

### A.5 PROOF OF CONDITIONAL INDEPENDENCE

Assume that two set of constraints are given, $C_1 = \{o_1, o_2, o_3, o_4\}$ and $C_2 = \{o_3, o_4, o_5, o_6\}$. In this section, we will show that $C_1$ and $C_2$ are conditional independent and hence the compositionality of our planner can be achieved as in (Liu et al., 2022). Define $\mathcal{J}_{c_1}$ and $\mathcal{J}_{c_2}$ as the set of trajectories that satisfy $C_1$ and $C_2$, respectively. Let $f_{C_i}(q_{1:T})$ denote a probabilistic density function of trajectories, whose value is 1 if $q_{1:T} \in \mathcal{J}$ and is 0 otherwise, that is

$$f_{C_i}(\mathcal{J}) = \begin{cases} 1 & \text{if } \mathcal{J} \in \mathcal{J}_{c_i} \\ 0 & \text{if } \mathcal{J} \notin \mathcal{J}_{c_i} \end{cases} \tag{19}$$

Then, note that for the joint probability density function for the union of the obstacles $f_{C_1 \cup C_2}$ is directly proportional to the product of the individual density functions (since set of trajectories satisfying both obstacles corresponds to $\mathcal{J}_{c_1} \cap \mathcal{J}_{c_2}$). Therefore, we have

$$f_{C_1 \cup C_2} \propto f_{C_1} f_{C_2}. \tag{20}$$

Hence, $C_1$ and $C_2$ are conditional independent.

