# OpenReview forum: "Potential Based Diffusion Motion Planning"
_ICLR.cc/2024/Conference — Submitted to ICLR 2024_

### Official Review · Reviewer_PtS9 · 2023-11-01

**Soundness:** 2 fair
**Presentation:** 2 fair
**Contribution:** 2 fair
**Rating:** 5
**Confidence:** 5

**Summary:**

The paper proposes a new approach to motion planning in high dimensional spaces using potential-based motion planning. Potential-based motion planning allows for the combination of different motion constraints by adding corresponding potentials, but it often suffers from local minima. The proposed approach involves training neural networks to learn easily optimizable potentials over motion planning trajectories, resulting in improved performance compared to classical and recent learned motion planning approaches.

**Strengths:**

The strengths of the paper are as follows:

1) They introduce a new approach to motion planning in high dimensional spaces using potential-based motion planning, which offers the advantage of composability by combining different motion constraints with corresponding potentials.

2) The proposed approach trains neural networks to learn easily optimizable potentials over motion planning trajectories, resulting in improved performance compared to classical and recent learned motion planning approaches.

3) The effectiveness of the approach is demonstrated by outperforming existing classical and learned motion planning algorithms. The approach showcases the composability of motion planning, allowing for the generalization to multiple sets of motion constraints and an increased number of objects.

4) The results and comparisons are extensive.

**Weaknesses:**

The weakness of the paper are as follows:

1) For higher dimensions, sampling based algorithms have proven to be the best choice. The authors have considered RRT* as the sampling based algorithm for comparison. Please note that RRT* is a old method. The authors must strive to compare their methods against recent state of the art approaches such as Informed RRT* [1], Fast Marching Tree (FMT*) [2], Batch Informed Trees (BIT*)[3], RABIT* [4] and ABIT* [5].

2) I worked on sampling based algorithms for a while now and the order of magnitude improvement from RRT* to ABIT* is 30 times more.

3) The authors must cite sufficient state of the art papers on sampling based motion planning and provide more convincing arguments as to why their approach must be preferred over these state of the art sampling techniques.

4) Furthermore, all the above mentioned algorithms have asymptotic optimality guarantees i.e. the cost of the feasible path converges to global optimal path in the limit of large number of samples.

5) In Fig. 4, I do not think it makes sense to measure the success rate for RRT* or other sampling based algorithms. This is because RRT* using collision checking modules and the RRT* is guaranteed to find a feasible path if it exists (proven in original RRT* paper). Since you are using NNs in your approach and other approaches under comparison, it makes sense to measure the success rate.

[1] Gammell, Jonathan D., Siddhartha S. Srinivasa, and Timothy D. Barfoot. "Informed RRT: Optimal sampling-based path planning focused via direct sampling of an admissible ellipsoidal heuristic." In 2014 IEEE/RSJ international conference on intelligent robots and systems, pp. 2997-3004. IEEE, 2014.

[2]Janson, Lucas, Edward Schmerling, Ashley Clark, and Marco Pavone. "Fast marching tree: A fast marching sampling-based method for optimal motion planning in many dimensions." The International journal of robotics research 34, no. 7 (2015): 883-921.

[3] Gammell, Jonathan D., Siddhartha S. Srinivasa, and Timothy D. Barfoot. "Batch informed trees (BIT*): Sampling-based optimal planning via the heuristically guided search of implicit random geometric graphs." In 2015 IEEE international conference on robotics and automation (ICRA), pp. 3067-3074. IEEE, 2015.

[4] Choudhury, Sanjiban, Jonathan D. Gammell, Timothy D. Barfoot, Siddhartha S. Srinivasa, and Sebastian Scherer. "Regionally accelerated batch informed trees (rabit*): A framework to integrate local information into optimal path planning." In 2016 IEEE International Conference on Robotics and Automation (ICRA), pp. 4207-4214. IEEE, 2016.

[5] Strub, Marlin P., and Jonathan D. Gammell. "Advanced BIT (ABIT): Sampling-based planning with advanced graph-search techniques." In 2020 IEEE International Conference on Robotics and Automation (ICRA), pp. 130-136. IEEE, 2020.

**Questions:**

1) A thorough literature review must be made for sampling based algorithms and convincing arguments on why your approach must stand out must be made in the introduction as well.

2) Comparisons with state of the art such as ABIT*, RABIT*, Informed RRT* must be made.

3) Do you have guarantees for your approach.

4) Since you are using NNs, there is no guarantee that the success rate would be 100%?

---

> ### Author Response · Authors · 2023-11-21
> **Response to Reviewer PtS9**
>
> Thank you for your constructive feedback. We have carefully considered each point in the review and updated the paper accordingly.
>
> > Q1. A more thorough literature review.
>
> Thank you for the suggestion. We have updated the Introduction section.
>
>
> > Q2. Comparisons with state of the art sampling-based methods.
>
> We have appended the results of BIT* in three environments: Maze2D, KUKA, Dual KUKA.
> In addition, we also added a potential-guided sampling-based method P-RRT* for reference.
> For a more complete quantitative table, please refer to the updated Table VIII in the appendix. Besides, We also added experiment results in another environment with 7 concave obstacles in the Table X.
>
> Maze 2D:
> | Method    | Success | Time  | Check     |
> |----------|----------|----------|----------|
> | RRT*      | 100.0  | 1.08  | 10561.73  |
> | P-RRT*      | 99.95  | 1.34  | 15697.09  |
> | BIT*      | 100.0   | 0.21  | 1894.20   |
> | Ours      | **100.0**   | **0.12**  | **49.07**     |
>
> KUKA 7D:
> | Method    | Success | Time  | Check     |
> |----------|----------|----------|----------|
> | RRT*      | 67.2  | 11.86 | 53282.03  |
> | P-RRT*      | 66.20 | 12.72 | 53590.14 |
> | BIT*      | **99.2**   | 1.17  | 7988.53   |
> | Ours      | **98.7**   | **0.13**  | **67.02**    |
>
> Dual KUKA 14D:
>
> | Method    | Success | Time  | Check     |
> |----------|----------|----------|----------|
> | RRT*      | 47.50  | 33.80 | 81759.71  |
> | P-RRT*    |  47.40 | 33.98  |104820.10 |
> | BIT*      |  94.95  | 4.60  |  20689.98  |
> | Ours      | **96.35**   | **0.30**  | **129.63**    |
>
> > Q3. Do you have guarantees for your approach.
>
> We have added the theoretical analysis of the guarantees of our approach in A.4 in the Appendix. Our approach is guaranteed to be probabilistically complete. The rough intuition is as follows. Since our learned neural network assigns positive likelihood to all trajectories, given a set of valid motion planning paths from A to B, our learned distribution will always assign finite positive likelihood to it. Thus, given a very large number of samples, our sampler is guaranteed to find a solution to the motion planning problem.
>
> Further, with optimal data and training, our approach can also be guaranteed to be asymptotically optimal. At a high level, since neural networks are universal function approximators, an optimal $E_\theta^*(.)$ can be learned by our models and the corresponding output distribution that can generate optimal plans almost surely.
>
>
> > Q4. Since you are using NNs, there is no guarantee that the success rate would be 100\%?
>
> As discussed earlier, our approach is also guaranteed to be probabilistically complete with a sufficient number of samples.
>
>
> > W5. Questions on measuring the success rate for RRT* or other sampling based algorithms.
>
> While with unlimited sampling time, both RRT* and BIT* are probabilistically guaranteed to converge, in practice, especially in settings with very narrow paths to solutions, the amount of to sample a valid path can take exponentially long (for instance if there is a very narrow passage way from side of the workspace to the other). Thus, we evaluate the performance of RRT* and BIT* with a fixed time limit.

---

> > ### Comment · Reviewer_PtS9 · 2023-11-22
> > **Response to authors**
> >
> > Thank you for the clarifications. I will be modifying my score.

---

> > > ### Author Response · Authors · 2023-11-22
> > > **Thank you for the score update**
> > >
> > > We are glad to read that our response is helpful. Thank you very much for the increased score.

---

### Official Review · Reviewer_i3W8 · 2023-11-01

**Soundness:** 2 fair
**Presentation:** 2 fair
**Contribution:** 2 fair
**Rating:** 5
**Confidence:** 3

**Summary:**

The paper provides a framework to adapt the diffusion model to the motion planning task. By considering the diffusion model as the potential-based energy model, the compositionality can be achieved by the addition of multiple independent energy models. Empirical results show that the learned model works well from 2D to 14D, and can generalize to unseen cases. Real-world datasets are also evaluated.

**Strengths:**

1. The paper is overall easy to understand.

2. The experiment looks promising.

**Weaknesses:**

1. Typos.
1a. Page 6, in the caption of Figure 4, there should be a space before '(b)'. And this caption text is not finished.
1b. Page 6, at the end of this page, 'spasrse' should be 'sparse'.

2. Though the compositionality seems to work well, the theoretical side is unclear. See question 2.

3. Some settings of the experiments need to be further clarified to evaluate the paper better. See question 3.

**Questions:**

1. About the formulation of the compositionality:

- 1a: It seems that the unconditioned score function (c=∅) occurs only in Line 7 of Algorithm 1. Equation 6, 7, 8 simply ignores this term and combines all the conditional scores. Which one is actually used?

- 1b: If Line 7 of Algorithm 1 is the one that is actually used, it differs from the papers (Equation 9 of [1]), in the sense that there is no subtraction by the unconditioned score. Why does the author choose such a form? Isn't this wrong if applying Bayes' theorem with Equation 10 from [2]?

- 1c: Still about Line 7 of Algorithm 1: Why does the unconditioned score use $E_{\theta}^1$? What is special about $E_{\theta}^1$ compared to the other $E_{\theta}^i$?

2. About the theory side of the compositionality:

- 2a: The basic assumption for Equation 10 from [2] to work is that all the conditions must be conditionally independent. Wouldn't this assumption be violated in this paper's experiment settings? For example, as a scenario mentioned in the paper, if C1={o1,o2,o3,o4} and C2={o3,o4,o5,o6}, will C1 and C2 still be conditional independent?

- 2b: Even the conditional independent assumption holds for t=T, for the intermediate t, is such an assumption still guaranteed, especially the distribution of $x_t$ now is actually affected by these conditions?

3. About the experiment settings:

- 3a: What is the representation of the dynamic obstacles? Is it a vector consisting of the configurations from all timesteps in the trajectory?

- 3b: Since the dataset is generated from BIT*, why not compare it as a baseline?

- 3c: A general impression of the diffusion model is that it takes a long time to generate samples. Why is the planning time here lower than the other baselines? Does the planning time also include the sampling time of the diffusion model and the finetune time? Is there any additional optimization you did to speed up the process?

- 3d. For the real-world dataset, is it possible to evaluate the model's success rate systematically (like overlap on pixels)? It is hard to parse the normal controller and the model's performances solely from images.

- 3e. What is the timeout condition for the planning?

- 3f. Are the metrics averaged over all the cases, or only the successful cases?

[1] Ajay, A., Du, Y., Gupta, A., Tenenbaum, J., Jaakkola, T., & Agrawal, P. (2022). Is conditional generative modeling all you need for decision-making?. arXiv preprint arXiv:2211.15657.

[2] Liu, N., Li, S., Du, Y., Torralba, A., & Tenenbaum, J. B. (2022, October). Compositional visual generation with composable diffusion models. In European Conference on Computer Vision (pp. 423-439). Cham: Springer Nature Switzerland.

---

> ### Author Response · Authors · 2023-11-20
> **Response to Reviewer i3W8 (1/2)**
>
> Thank you very much for carefully reading the paper and the valuable insights. We have added experiment results and updated the manuscript accordingly.
>
>
> > Q1. About the formulation of the compositionality.
>
> (1a, 1b): Confusion about the actual equations used in the denoising process.
> \
> Sorry for the confusion caused.
> The equation we used is the Line 7 of Algorithm 1 and the unconditioned score is also used. Thus, the full equation in Line 7 is given by:
>
> \begin{align*}
>     \epsilon_{\text{comb}} =
>       \nabla_{q_{1:T}} E_\theta(q_{1:T}^s, s, q_{\text{start}}, q_{\text{end}}, \varnothing) +
>        \sum_{i=1}^{N} \omega_i \nabla_{q_{1:T}} \big[ E_\theta^{i}(q_{1:T}^s, s, q_{\text{start}}, q_{\text{end}}, C_i) - E_\theta^{i}(q_{1:T}^s, s, q_{\text{start}}, q_{\text{end}}, \varnothing) \big]
> \end{align*}
>
> Likewise, in equation 5, the $\nabla_{q_{1:T}} E_\theta(q_{1:T}, t, q_{\text{start}}, q_{\text{end}}, C)$ should be
>
> \begin{align}
>      & \nabla_{q_{1:T}} E_\theta(q_{1:T}, t, q_{\text{start}}, q_{\text{end}}, \varnothing) - \omega \nabla_{q_{1:T}}
>      \big[
>        E_\theta(q_{1:T}, t, q_{\text{start}}, q_{\text{end}}, C)  -
>        E_\theta(q_{1:T}, t, q_{\text{start}}, q_{\text{end}}, \varnothing)
>      \big]
> \end{align}
>
>
> For simplicity, we would like to use abbreviated forms in the original submission to stand from the update gradient but miss a proper declaration in the paper. We have updated the equations in the manuscript.
> \
> \
> \
> (1c): About Line 7 of Algorithm 1: Why does the unconditioned score use $E_\theta^1$? What is special about $E_\theta^1$ compared to the other $E_\theta^i$?
> \
> In practice, we need an unconditional score, though the particular model we get from is not important. In our experiments, we used $E_\theta^1$, but to make this clear we have replaced $E_\theta^1$ with $E_\theta$ in the algorithm block. The unconditional score can provide a basic prior of a motion trajectory, such as the coherency between two consecutive waypoints and kinematics constraints, while the conditioned score can provide environment specific constraints.
>
>
> > Q2. About the theory side of the compositionality:
>
> (2a): If C1={o1,o2,o3,o4} and C2={o3,o4,o5,o6}, will C1 and C2 still be conditional independent?
> \
> We provided a detailed proof in the updated paper appendix A.5.  At a high level, if we view the distribution p(traj|C1) as a uniform density over all trajectories avoiding C1 and p(traj|C2) as a uniform density over all trajectories avoiding C2, then p(traj|C1 $\cap$ C2) is directly proportional to the product of p(traj|C1) and p(traj|C2) and is conditionally independent.
> \
> \
> (2b): Is the assumption of conditional independence still guaranteed for the intermediate t?
> \
> It's true that even if the independence holds at t=T, for the intermediate t, the score functions are not fully independent (due to added noise). However, as discussed in [1], even if the intermediate scores are not fully accurate, as long as they transition to 2 independent distributions at the end, the sampling is still theoretically sound.
>
> [1] Du, Yilun, et al. "Reduce, reuse, recycle: Compositional generation with energy-based diffusion models and mcmc." International Conference on Machine Learning. PMLR, 2023.

---

> ### Author Response · Authors · 2023-11-21
> **Response to Reviewer i3W8 (2/2)**
>
> > Q3. About the experiment settings.
>
> (3a): What is the representation of the dynamic obstacles?  Is it a vector consisting of the configurations from all timesteps in the trajectory?
> \
> Yes. In our setting, the representation of dynamic obstacles is a sequence of configurations indicating the trajectories of the presented obstacles. This representation is aligned with our baseline method SIPP.
> \
> \
> (3b): Comparison to BIT*.
> \
> We have enclosed the performance of BIT*. The timeout condition for planning is set to 30 seconds for BIT*.
> We set the timeout condition of RRT* to 30 seconds in Maze2D and KUKA7D, and 60 seconds in Dual KUKA 14D.
>
> Maze 2D:
> | Method    | Success | Time  | Check     |
> |----------|----------|----------|----------|
> | RRT*      | 100.0  | 1.08  | 10561.73  |
> | BIT*      | 100.0   | 0.21  | 1894.20   |
> | Ours      | **100.0**   | **0.12**  | **49.07**     |
>
> KUKA 7D:
> | Method    | Success | Time  | Check     |
> |----------|----------|----------|----------|
> | RRT*      | 67.2  | 11.86 | 53282.03  |
> | BIT*      | **99.2**   | 1.17  | 7988.53   |
> | Ours      | **98.7**   | **0.13**  | **67.02**    |
>
> Dual KUKA 14D:
>
> | Method    | Success | Time  | Check     |
> |----------|----------|----------|----------|
> | RRT*      | 47.50  | 33.80 | 81759.71  |
> | BIT*      |  95.0  | 4.60  |  20689.98  |
> | Ours      | **96.4**   | **0.30**  | **129.63**    |
>
> (3c): A general impression of the diffusion model is that it takes a long time to generate samples. Why is the planning time here lower than the other baselines?
> \
> The main speed-up is due to the use of DDIM[1] to accelerate the denoising process, which can reduce the number of denoising steps compared to DDPM. We have included the DDIM sampling details in Appendix A.2.3.
> \
> (3d): Quantitative results on the real-world dataset.
> \
> Thank you for the suggestion. We have appended quantatitive results on real-world ETH/UCY dataset in Table IX of A.3.1 in the appendix.
> We found using pixels overlap might be inaccurate due to the volume of pedestrians and their various moving dynamics.
> Therefore, we directly compare the similarity between the predicted motion trajectories and ground truth human trajectories in a held-out test set. We use ADE as the metric. Please refer to the A.3.1 in the Appendix for more details.
>
>
> |   |   ETH  |  |  Hotel |   | Zara01  |   | Zara02  |   | Students01  |   | Students03  |   |
> |--------|--------|---|---|---|---|---|---|---|---|---|---|---|
> | Method | ADE↓ | Time | ADE | Time | ADE | Time | ADE | Time | ADE | Time | ADE | Time |
> | MPNet | 18.11 | 0.12 | 28.60 | 0.11 | 11.06 | 0.12 | 17.22 | 0.11 | 10.37 | 0.12 | 8.93 | 0.11 |
> | MπNet | 37.70 | 0.26 | 44.49 | 0.29 | 1.14 | 0.22 | 13.66 | 0.23 | 12.76 | 0.18 | 1.54 | 0.22 |
> | Ours | **0.94** | **0.17** | **5.20** | **0.17** | **0.35** | **0.17** | **0.38** | **0.17** | **0.52** | **0.17** | **0.89** | **0.17** |
>
>
> (3e): What is the timeout condition for the planning?
> \
> For sampling-based methods, the timeout condition of BIT* is set to 30 seconds across all environments. For RRT* and P-RRT*, the timeout condition is 60 seconds in Dual KUKA 14D, and 30 seconds in all other environments.
> For MPNet, we directly adopt the default number of trials, 80, as given their codebase. That is, a timeout flag will raise if the model has been called 80 times and still had not reached the target. Likewise, we set the number of trials to 150 for M$\pi$Net and AMP-LS.
> \
> \
> (3f): Are the metrics averaged over all the cases, or only the successful cases?
> \
> The metrics are averaged over all the test cases.
> \
> \
> [1] Song, Jiaming, Chenlin Meng, and Stefano Ermon. "Denoising Diffusion Implicit Models." International Conference on Learning Representations. 2020.

---

> > ### Comment · Reviewer_i3W8 · 2023-11-22
> > **Thank you for your reply**
> >
> > Thank you for your great effort. I think most of my concerns are addressed. Typically, I see:
> > - the revision to the compositionality equation
> > - the explanation for the conditional independence in the intermediate timestep (need MCMC to fix it, but still doable)
> > - the comparision to bit* and p-rrt*
> > - the quantitative results on real-world dataset (still a metric related to the collision would be appreciated, since we are doing motion planning instead of pure imitation learning. I would imagine a low ADE method will also yield a higher success rate, but this claim probably still needs to be validated by numerical results)
> >
> > However, the issue of conditional independence (and consequently, the optimality for composition) still remains. The proof given in the appendix looks like a circular reasoning and is fundamentally wrong. I will give a counter-example here. Imagine there are 10 trajectories in the whole trajectory space, and each trajectory can or cannot satisfy {o1,o2,o3,o4} and {o3,o4,o5,o6}. The following table describes such a situation:
> >
> > |     |{o1,o2,o3,o4} is violated | {o1,o2,o3,o4} is satisfied |
> > |-----|-----|-----|
> > |**{o3,o4,o5,o6} is violated** | {$J_1, J_2$} | {$J_3, J_4, J_5$} |
> > | **{o3,o4,o5,o6} is satisfied** | {$J_6$} | {$J_7, J_8, J_9, J_{10}$} |
> >
> > This is not a conditional independence case, since:
> >
> > > p(constraints are satisfied | {o1,o2,o3,o4, o5, o6})= 0.4
> > >
> > > p(constrains are satisfied | {o1,o2,o3,o4}) $\cdot$ p(constrains are satisfied | {o3,o4, o5, o6}) = 0.7 * 0.5 = 0.35
> >
> > Clearly, such a case could happen in real-world applications. Imagine for $J_3, J_4, J_5$, these trajectories try to avoid {o1,o2,o3,o4}, but run into the direction where {o5,o6} are, which results as collisions. It is hard to say that similar cases would not occur in this paper's experiments without further evidence.
> >
> > -------------------------------
> >
> > Instead of testing whether the conditional independence is valid in this paper's applications using density estimation methods, I would suggest the author:
> >
> > 1. Remove the current proof for optimality and conditional independence. These are factual errors and fundamentally wrong. An accepted paper should not include factual errors. Universal approximation only works when the training and test distribution share the same one, and do not account for the compositionality. When the test distribution becomes the composite distribution, the universal approximation cannot guarantee the behavior for the compositionality.
> >
> > 2. Provide more discussions about conditional independence for both the terminal and the intermediate timesteps in the limitation section. For example, "This paper does assume conditional independence, which can often be wrong in real-world applications, and the composition equation might not work when the constraints are correlated. But as we can see, the method often works empirically well in these imperfect cases."
> >
> >      An accepted paper does not need to be perfect and have no drawbacks, but it needs to show what the limitations are and what assumption is needed. Providing more information about the limitations helps the community understand better when your method is applicable and how your method could fail in the long term.

---

> ### Author Response · Authors · 2023-11-23
> **Response to Reviewer i3W8's reply**
>
> Thank you for the patience and detailed illustration of your concerns.
> In our proof in Appendix A.5,
> We would like to emphasize that the sampling procedure at inference time actually won't be affected even in the case that $C_1 = \{o_1, o_2, o_3, o_4\}$ and $C_2 = \{ o_3, o_4, o_5, o_6 \}$.
>
> Our previous response might cause your concerns and following is a more thorough explanation.
> We define $f_{C_i}(q_{1:T})$ as the probability density function over trajectories, where we uniformly assign positive likelihood to the trajectory $q_{1:T}$ if it satisfies the constraint $C_i$; otherwise, the likelihood is set to 0. Let $\mathcal{J}_{c_i}$ denote the set of trajectories that satisfy $C_i$. Then, we have
>
> \begin{equation}
>     f_{C_i}(q_{1:T}) =   \rho_i  \text{ if } q_{1:T} \in \mathcal{J}, \quad  0 \text{ if } q_{1:T} \notin \mathcal{J}_{c_i}
> \end{equation}
>
> where $\rho_i$ is a small constant (the first $\mathcal{J}$ should also have subscript $c_i$, we cannot render the whole equation if we put the $c_i$ in place). Similarly, we can define $f_{C_1 \cup C_2}$ as the probability density function of trajectories that satisfy both $C_1$ and $C_2$. Clearly, given any trajectory $q_{1:T}$, the probability density of $q_{1:T}$ is positive if and only if both $f_{C_1}(q_{1:T})$ and $f_{C_2}(q_{1:T})$ are positive. More specifically, we have
> \begin{equation}
>     f_{C_1 \cup C_2}(q_{1:T}) = \gamma f_{C_1}({q_{1:T}}) f_{C_2}({q_{1:T}})
> \end{equation}
>
> where $\gamma$ is a constant (note the proportionality constant in the previous equation). We can see that the joint probability density function equals to the scaled product of $f_{C_1}$ and $f_{C_2}$. While the above equation doesn't indicate independence, sampling using the score function for left side of the equation is the same as sampling from the summed score function for the right side of the equation because the score function is the gradient of the log probability and is invariant to the constant multiplier. Thus, the constant $\gamma$ here will not affect the test-time sampling process and thus, the procedure is correct.

---

> > ### Comment · Reviewer_i3W8 · 2023-11-23
> > **Thank you for your reply**
> >
> > Thanks. A critical assumption here is that the feasible trajectories are uniformly distributed in the trajectory space, and I still question whether such an assumption is valid in high-dimensional or nonlinear motion planning tasks. Usually, there should be some region in the space that can have a higher possibility to produce feasible trajectories than the other region. Even though you do uniform sampling during training, there should still exist some trajectories that tend to concentrate together in the dataset, which makes the distribution non-uniform.
> >
> > A modified example is as follows:
> >
> > |     |{o1,o2,o3,o4} is violated | {o1,o2,o3,o4} is satisfied |
> > |-----|-----|-----|
> > |**{o3,o4,o5,o6} is violated** | {$J_1, J_2$} | {$J_3, J_4, J_5$} |
> > | **{o3,o4,o5,o6} is satisfied** | {} | {$J_7, J_7, J_8, J_9, J_{10}$} |
> >
> > Then:
> >
> > >p($J_7$ | {o1, o2, o3,o4,o5,o6}) /  (p($J_7$ | {o1,o2,o3,o4}) $\cdot$ p($J_7$ | {o3,o4, o5, o6}))  = 4
> > >
> > >p($J_8$ | {o1, o2, o3,o4,o5,o6}) /  (p($J_8$ | {o1,o2,o3,o4}) $\cdot$ p($J_8$ | {o3,o4, o5, o6}))  = 8
> >
> > which clearly shows that $\gamma$ is no more a constant.
> >
> > I still recommend the author to discuss this issue in the limitation section.

---

### Official Review · Reviewer_tz1K · 2023-11-03

**Soundness:** 1 poor
**Presentation:** 2 fair
**Contribution:** 1 poor
**Rating:** 1
**Confidence:** 5

**Summary:**

The paper proposes a potential field motion planning approach which uses score matching to learn energy functions which represent the attractor and repulsor potentials. The potentials are then combined via simple summation, producing a velocity policy where trajectories can be rolled-out. The method is pretty straightforward, and the paper easy to follow. The paper demonstrates the applicability of the method on pedagogical examples.

My main concerns with the paper is as follows:

1. The introduction states that potential field methods have "fallen out of favour in recent years", this is not quite true, as reactive motion generation approaches, notably Riemannian Motion Policies (RMPs) (Ratliff, 2018) and Geometric Fabrics (GF) (van Wyk, 2022) are in effect potential field methods. RMPs and GFs are highly sophisticated frameworks, capable of composing potentials defined on multiple task-spaces, and combine each potential according to a weighting metric. This is in contrast to the simplistic approach proposed in this paper, which simply considers potentials in the same task space, and then naively sum everything together.

2. Score-matching learns energy functions where the parameterised function's gradients match the target function, however, the energy function values themselves can be very different. This makes directly summing the potentials very unsound. What if the energy values of one obstacle is higher than that representing another obstacle?

3. SDFs are ubiquitous in robotics, and can be thought of as a potential that increases as one moves away from the surface of the obstacle. What would be the motivations of learning a potential to represent the obstacle when ones could build an SDF, which can model complex scenes very efficiently, and use that as the repulsor?

4. The obstacle potential should not be in the C-space, it would be much easier to be constructed in the workspace, as it depends on the geometry of the workspace. However, the attractor potential is defined in the C-space. It is unclear how you combined these via simple addition.

**Strengths:**

See above.

**Weaknesses:**

See above.

**Questions:**

See above.

---

> ### Author Response · Authors · 2023-11-20
> **Response to Reviewer tz1K**
>
> Thank you for your constructive feedback. We have taken your comments into careful consideration and made revisions to our paper accordingly.
>
> > Q1. Missing some references.
>
> We apologize for missing the references to RMPs and GF potential field methods. We have updated our Introdution and added the references in the Related Work.  However, we believe that the contributions of our work is very different than these works, as we aim to learn potential fields generically from data for any types of potentials, including ones without clear close-form solutions such as multi-agent interaction. As potentials are learned from data, we found that our approach is much less vulnerable to adversarial minima compare to RMP (see newly updated Table X and Figure IX in the appendix our updated paper). Our approach could also be applied to different task spaces, by using the Jacobian to transform the inputs to each learned potential function into a shared task space.
>
>
> > Q2. Concerns on directly summing the potentials.
>
> Empirically, we found that this worked well as demonstrated in our quantitative results in Table 3 and Table XI, and the inferred energy functions have a naturally defined scale based off the learned probability density over demonstrations. In cases when this is inadequate, we can use failures in motion planning to help tune the right coefficient to add to energy functions.
>
>
>
> > Q3. Usage of SDF as the repulsor.
>
> SDF does provide accurate geometry to guide potential-based motion planning methods in the corresponding workspace.
> However,
>
> 1. simply using a SDF might not be able to resolve the local minima issue for the potential-based methods.
> 2. Constructing a SDF is highly workspace and robot dependent, and requires sophisticated computatation framework, which is especially challenging given limited observations of the environment.
>
>
> > Q4: The obstacle potential should not be in the C-space, it would be much easier to be constructed in the workspace.
>
> As we directly learn potentials for both obstacles and attractors, we can define our learned potentials in any space. In our experiments, we directly learn our potentials in C-space, allowing them to be directly combined via simple addition.

---

### Official Review · Reviewer_jxc2 · 2023-11-07

**Soundness:** 3 good
**Presentation:** 3 good
**Contribution:** 2 fair
**Rating:** 5
**Confidence:** 4

**Summary:**

This paper seeks to address the problem of motion planning using a novel potential based method that leverages diffusion models. The main contribution lies in the proposed compositional potential based diffusion motion planning with motion plan refinement, as well as the experiment comparison against multiple baseline algorithms including evaluations on a real-world dataset.

**Strengths:**

+ The idea of leveraging recent advance in diffusion models for potential based motion planning is interesting.

+ The paper is well written in general that clearly presents the basic idea and how the algorithm works.

+ Experiments on simulation and real-world dataset are provided to demonstrate the effectiveness of the proposed method.

**Weaknesses:**

- The literature review of motion planning is quite substandard. Authors are strongly encouraged to discuss the comparison against reactive local planning methods with collision avoidance such as velocity obstacles and safety barrier certificates.

- While the idea of using diffusion model is interesting, the paper fails to justify how the introduction of diffusion model could overcome local minima issues suffered from traditional potential field based approaches. In fact, all the static obstacles in the provided environment examples in Fig. 1-3 and Fig. 7 are convex and without any overlaps, where local minima may not exist even if using traditional potential field based planning.

- According to Algorithm 1, it seems the presented diffusion motion planning is a single-query planning technique that would require re-training for every different pair of start and goal configurations, which raises concern about the computation efficiency.

- It is unclear whether the presented planning method has any guarantees or empirical analysis in terms of collision avoidance and completeness. For instance, how to prove the denoising process in Eq. 10 does not introduce potential collisions of the new plan?

**Questions:**

Besides the items discussed above (see "weakness"), please find the additional questions in the following:

1. Could authors provide additional results showing the planning performance of the diffusion-based approach in environments with concave obstacles? With the given examples in Fig. 1-3 and Fig. 7 where only convex obstacles are presented, it is difficult to evaluate the improvement over local minima compared to traditional potential field based planning.

2. Could authors provide formal theoretical analysis to justify the presented method is free of collisions and comment on the optimality and completeness? For example, it seems the denoising process from Eq. 10 has no guarantees on collision avoidance for the new path.

3. Does the proposed diffusion motion planning need to be re-trained for every single pair of start and goal configurations? Could the algorithm be used for kinodynamic planning where the way two consecutive waypoints connect is constrained due to the kinematics and dynamic constraints of the robot model?

---

> ### Author Response · Authors · 2023-11-20
> **Response to Reviewer jxc2 (1/2)**
>
> Thank you for your insightful review and feedback. We have updated our manuscript and added experiment results based on your comments.
>
> > W1. Literature review and comparison with reactive local planning methods.
>
> We have updated Introduction and Related Work of the paper and added the discussion on the mentioned methods.
> Our approach is very different from the traditional reactive motion planning approaches in that:
>
> 1. Our method is a learning-based motion planning approach that directly generates an end-to-end trajectory conditioned on the robot state state, goal state, and global environmental geometry. In comparison, reactive local planning methods usually adopt a greedy strategy and focus on local geometry, which is prone to be stuck in the local minima.
> 2. Purely reactive planning methods typically need to construct implicit obstacle representation in the robot configuration space, which demands sophisticated computational frameworks or even might require case-by-case designs. By contrast, our model is trained directly on the demonstrations, which is more accessible especially in high-dimensional robot configuration space since the  potential of obstacles is learned in the robot configuration space.
>
>
>
> > W2. How our methods can overcome the local minima issues.
>
> Convex obstacles can also result in the local minima issue in traditional potential-based planning. For example, the agent might get stuck due to the balance of the attractive force and repulsive force.
>
> We summarize the reasons why the proposed planner can effectively avoid local minima below:
>
> 1) **Planning with the learned global potential field.** Our method generates motion plans by effectively considering the global geometry, start pose, and goal pose holistically in the latent space. The learned potential field can provide a strong global prior in long-horizon planning, in contrast to traditional potential-based methods that primarily consider the obstacles in the vicinity of the current pose.
> 2) **Stochasticity in motion plan generation.** The denoising process of our potential-based diffusion model is inherently stochastic, lowering the risk of local minima. Even when the the planner is stuck in a local minima, new plans can be generated by starting denoising from another noise, or by refining the problematic motion plans as given in Section 3.4. By contrast, most traditional potential based methods are deterministic and with limited capability to avoid or recover from the local minima.
> 3) **Adaptive weighting in the latent space.** Traditional potential-based methods usually comprise of attraction forces and repulsion forces and are sensitive to their weights. Such weights are highly problem dependent and a bad set of weights is one important reason for the local minima issues. In comparison, our learned potential method maps these two forces together in the unified learned latent space. Thus, the weighting is implicitly performed in the latent space based on the given conditions, which also avoids the heavy weight-tuning tasks.
>
> > W3&Q3: Is our method a single-query planning technique that would require re-training?
>
> Our method does not need to be re-trained and can be generalized to any valid start and goal configuration. In test time, the input is the start state, goal state, and the environment geometry (e.g., obstacles information). No parameters of the model need to be updated.
>
> Our model directly learns from the valid demonstration trajectories  with no violation of kinematics and dynamic constraints of a given robot model. Such physically improbable trajectories are out-of-distribution samples and contradicts to the model's output distribution. If there is any newly added test-time dynamic constraints, we believe that our method can further subject to them by composing the corresponding potentials during motion generation.
>
>
>
> > W4&Q2: Proof of probabilistic completeness and optimality.
>
> We give a proof on the probabilistic completeness and optimality in Appendix A.4. At a high level, since our learned neural network assigns positive likelihood to all trajectories, given a set of valid motion planning paths from A to B, our learned distribution will always assign finite positive likelihood to it. Thus, given a very large number of samples, our sampler is guaranteed to find a solution to the motion planning problem.
>
> Finding the optimal $E_\theta^*(.)$ for asymptotic optimality, however, is hard. Empirically, we found that the optimization is easy, robust, and can produce decent trajectories, as shown in Table VIII, IX, X, XI, and Figure 6, 7, IX, X. Moreover, as for speed, it is sufficient for our method to generate high-quality motion trajectories with less than 10 optimization steps across many different environments.

---

> ### Author Response · Authors · 2023-11-20
> **Response to Reviewer jxc2 (2/2)**
>
> > Q1. Additional results on environments with concave obstacles and  potential-based methods.
>
> We constructed an additional environment with 7 concave obstacles and the quantitative results are shown below. Besides, we added a traditional potential-based motion planning method RMP[1] and a potential-guided sampling-based method P-RRT*[2].
> We have also appended the qualitative results in Figure IX in the updated paper. For a more completethe quantitative table, please refer to the updated Table VIII and Table X in the appendix.
>
>
> **Motion Planning Performance on Maze2D with 7 Concave Obstacles:**
> | Method    | Success | Time  | Check     |
> |----------|----------|----------|----------|
> | RRT*      | 100.0  | 2.53  | 12601.23  |
> | P-RRT*    | 99.9   | 3.31  | 17076.69  |
> | BIT*      | 100.0  | 0.45  | 2251.96   |
> | MPNet     | 84.3   | 0.38  | 2265.30   |
> | M$\pi$Net | 98.7   | **0.06**  | 67.60      |
> | RMP       | 28.0   | 0.34  | --        |
> | Ours      | **100.0**  | **0.15**  | **52.10**      |
>
>
> **Additional results of P-RRT:***
>
> Maze 2D (convex):
> | Method    | Success | Time  | Check     |
> |----------|----------|----------|----------|
> | P-RRT*      | 99.95  | 1.34  | 15697.09  |
> | Ours      | **100.00**   | **0.12**  | **49.07**     |
>
>
> KUKA 7D:
>
> | Method    | Success | Time  | Check     |
> |----------|----------|----------|----------|
> | P-RRT*      | 66.20 | 12.72 | 53590.14 |
> | Ours      | **98.65**   | **0.13**  | **67.02**    |
>
>
> Dual KUKA 14D:
>
> | Method    | Success | Time  | Check     |
> |----------|----------|----------|----------|
> | P-RRT*    |  47.40 | 33.98  |104820.10 |
> | Ours      | **96.35**   | **0.30**  | **129.63**    |
>
>
> [1] Ratliff, Nathan D., et al. "Riemannian motion policies." arXiv preprint arXiv:1801.02854 (2018).
>
> [2] Qureshi, Ahmed Hussain, and Yasar Ayaz. "Potential functions based sampling heuristic for optimal path planning." Autonomous Robots 40 (2016): 1079-1093.

---

> > ### Comment · Reviewer_jxc2 · 2023-11-23
> > **Thank you for the additional information**
> >
> > Thanks authors for the efforts in clarifying some of my concerns, and I do agree with the mentioned difference about how the proposed method works compared to the conventional potential-based methods. However, the proposed method still lacks fundamental guarantees on the promised properties, such as free from local minima, etc. In particular, having an end-to-end design does not essentially justify the performance of the algorithm. To that end, I will keep my score as is.

---

> > > ### Author Response · Authors · 2023-11-23
> > > **Thank you for reviewing**
> > >
> > > Thank you again for your time and effort in reviewing our work.

---

### Author Response · Authors · 2023-11-21
**General Updates for Rebuttal**

We thank all reviewers and ACs for their time and effort in reviewing our paper and for their constructive comments.

We are glad to learn that the reviewers find the following contributions of our work:

**Method**: The idea of leveraging the potential-based diffusion model in motion planning is novel and interesting (jxc2).

**Experiments:** Our results and comparisons across multiple environments and real-world datasets are comprehensive and promising (jxc2, i3W8, PtS9).

**Presentation:** The paper is well-written and clearly presents the proposed algorithm (jxc2). Paper is overall easy-to-follow (i3W8).

We have replied to the reviewers' concerns in the individual responses. We also revised our manuscript following the reviewers' suggestions. We would like to note that we made the following major updates according to the suggestions by the reviewers:
1. We have added **experiment results on a new Maze2D environment with concave obstacles**, as shown in Table X, Figure IX, and the response to jxc2.

2. We have added **experiment results of three methods**: traditional potential-based method RMP, potential guided sampling-based method P-RRT*, and a more advanced sampling-based method BIT*. Please refer to Table VIII and Table X in the manuscript.

3. We have added the **quantitative results on real-world dataset**. Please refer to Appendix A.3.1 and Table IX for more details.

4. We have given **a proof on the probabilistic completeness and optimality** of our method in Appendix A.4.

5. We have **updated the introduction and related work** in the manuscript with a more thorough comparison with the existing methods.

---

### Meta-Review · Area_Chair_NVpo · 2023-12-06

**Metareview:**

Summary: This paper focuses on potential-field based motion planning. To generate motion plans, this paper uses a diffusion model representation to learn these trajectories and composes several motion planning constraints via additively multiple diffusion energy functions.

Strengths: The paper is intuitive and easy to follow. The results and comparisons are extensive, evaluating both on high-dimensional robot manipulator motion and on real-world pedestrian prediction (e.g., ETH).

Weaknesses: The main concern that all reviewers raised was over the claim that the diffusion model can escape the local minima issues typically faced by potential field based motion planners. Reviewers also suggested stronger baselines grounded in sampling-based motion planning (e.g., BIT*), potential-field literature (e.g. Velocity Obstacles), and alternative approaches (e.g., signed distance functions). The results indicate that BIT* results are very close to the proposed method. In future versions of the manuscript, I suggest reporting mean and standard deviation of success over multiple initial conditions to understand if the differences in the methods are statistically significant.

**Justification For Why Not Higher Score:**

After I carefully reviewed the manuscript and the author-reviewer discussion, I recommend rejection at this time. I encourage the authors to perform further comparisons on the proposed approach vs. SOTA random motion planning and potential-based planning, as well as expanding on the theoretical completeness and optimality results (which are currently in the appendix text) within the main text.

**Justification For Why Not Lower Score:**

N/A

---

### Decision · Program_Chairs · 2024-01-16

Reject